# Rab30 facilitates lipid homeostasis during fasting

Danielle M. Smith[1,2], Brian Y. Liu [1] & Michael J. Wolfgang [1,2,3] ✉

To facilitate inter-tissue communication and the exchange of proteins, lipoproteins, and metabolites with the circulation, hepatocytes have an intricate and efficient intracellular trafficking system regulated by small Rab GTPases. Here, we show that Rab30 is induced in the mouse liver by fasting, which is amplified in liver-specific carnitine palmitoyltransferase 2 knockout mice (Cpt2[L−/−]) lacking the ability to oxidize fatty acids, in a Pparα-dependent manner. Live-cell super-resolution imaging and in vivo proximity labeling demonstrates that Rab30-marked vesicles are highly dynamic and interact with proteins throughout the secretory pathway. Rab30 whole-body, liver-specific, and Rab30; Cpt2 liver-specific double knockout (DKO) mice are viable with intact Golgi ultrastructure, although Rab30 deficiency in DKO mice suppresses the serum dyslipidemia observed in Cpt2[L−/−] mice. Corresponding with decreased serum triglyceride and cholesterol levels, DKO mice exhibit decreased circulating but not hepatic ApoA4 protein, indicative of a trafficking defect. Together, these data suggest a role for Rab30 in the selective sorting of lipoproteins to influence hepatocyte and circulating triglyceride levels, particularly during times of excessive lipid burden.

The liver plays a crucial role in maintaining systemic glucose and lipid homeostasis in the face of an ever-changing nutritional environment. Hepatocytes, highly polarized epithelial cells, are sensitive to changes in nutrient availability and exhibit an efficient intracellular trafficking system to enable the storage, packaging, and transport of macronutrients in response to a dynamic metabolic environment[1]. The fed-to-fast transition is an example of when there is a high demand placed on the hepatocyte to both maintain blood glucose levels and to supply peripheral tissues with energetic substrates for survival. In the fasted state, there is a shift in substrate utilization from carbohydrates to lipids. This transition has been well-characterized in the liver and is marked by increased mitochondrial fatty acid β-oxidation, gluconeogenesis, and ketogenesis. Much of the lipid catabolic transcriptional program is governed by activation of the nuclear hormone receptor peroxisome proliferator activated receptor α (Pparα) through increased liganding of bioactive fatty acids derived from fasting-induced adipose tissue lipolysis[2]. Besides inducing transcription of

genes involved in lipid catabolism, Pparα activation has also been suggested to influence overall hepatic energy balance by promoting autophagy, suppressing energy-costly protein secretory processes, and affecting serum triglyceride- and cholesterol-rich lipoprotein levels[3–7].

Their robust intracellular trafficking system allows hepatocytes to vectorially exchange factors with the serum, recycle ligand-bound plasma membrane receptors, and transport cargo across several apical and basolateral domains with high fidelity. Due to this necessity for efficient endocytosis and secretion, the family of Rab GTPases play a particularly important role in polarized epithelial cells such as hepatocytes. Rabs are the principal regulators of membrane identity and control intracellular traffic by recruiting specific effectors to carry out vesicular transport and other membrane fusion events[8]. While the fission yeast S. pombe has 7 Rab family members, mammals have at least 60 identified Rab paralogs, implicating a correlation between the number of Rab proteins and multicellularity and cell specialization in

[1]Department of Physiology, The Johns Hopkins University School of Medicine, Baltimore, MD 21205, USA. [2]Department of Biological Chemistry, The Johns Hopkins University School of Medicine, Baltimore, MD 21205, USA. [3]Department of Pharmacology and Molecular Sciences, The Johns Hopkins University School of Medicine, Baltimore, MD 21205, USA. ✉e-mail: mwolfga1@jhmi.edu

animals[9–11]. Indeed, with the increase in tissue diversification and cellular identity comes higher demand for cooperation amongst different cell types to coordinate a uniform response to environmental changes. However, the physiological roles of many Rabs in animals have yet to be characterized.

Rab30 is one such metazoan-restricted Rab family member with an enigmatic function, especially in regards to liver physiology. Rab30 expression was found to be significantly induced in the liver during the priming phase of regeneration after partial hepatectomy, and it is the only member of the Rab family to exhibit such regulation; along with the observation that Rab30 expression exhibits significant upregulation in the livers of newborn mice, it was proposed that Rab30 could aid in a functional switch from hematopoietic support to metabolic homeostasis[12]. More evidence for the role of Rab30 in the maintenance of liver function comes from research revealing that Rab30 is among the most dysregulated genes early in the pathogenesis of drug-induced liver injury and correlates with later adverse histopathology events in rodents[13]. Furthermore, others have observed that Rab30 expression is induced in the liver in the fasting state and is blunted in Pparα knockout mouse models[14–17]. Despite these intriguing observations and associations of Rab30 and its regulation in the liver under metabolic stress, its role and requirement in liver function is not clear.

In this study, we define a physiological role for Rab30 in hepatic lipid and protein trafficking during fasting and in pathologically fatty livers. We find that Rab30 expression is specifically upregulated in the mouse liver during fasting in a Pparα-dependent manner. In live hepatocytes, Rab30 dynamically associates with the Golgi and marks post-Golgi membranes. Proximity labeling of interactors in live mice revealed that Rab30 interacts with proteins that localize to all parts of the secretory pathway, including the plasma membrane. Rab30 knockout animals are viable and fertile with normal Golgi apparatus morphology. Proteomics of 24 h fasted livers reveals that the loss of Rab30 on the Cpt2$^{L-/-}$ background (Rab30$^{L-/-}$; Cpt2$^{L-/-}$ double knockouts, or DKO) results in a significant retention of hepatocyte secreted proteins compared to Cpt2$^{L-/-}$ that are unaltered on the transcriptional level between the two genotypes, indicating a defect in protein secretion and/or turnover. The loss of Rab30 on the Cpt2$^{L-/-}$ background also reverses serum dyslipidemia and hypercholesterolemia in the Cpt2$^{L-/-}$ knockout mice. Corresponding to decreased serum triglyceride levels, we observe a decrease in serum apolipoprotein A4 (ApoA4) abundance in fasted DKO mice. Taken together, our results demonstrate a role for Rab30 in regulating protein and lipid trafficking during fasting in the hepatocyte.

## Results

### *Rab30* is induced in the liver by fasting and requires Pparα
Previously, we showed that the liver specific loss of carnitine palmitoyltransferase 2 (Cpt2$^{L-/-}$) and therefore hepatic mitochondrial long chain fatty acid β-oxidation resulted in dramatic fasting induced changes in the liver transcriptome and proteome[18]. In addition to changes in well characterized lipid metabolic genes, a relatively uncharacterized Rab GTPase, Rab30, was one of the most highly upregulated transcripts and proteins in Cpt2$^{L-/-}$ livers upon fasting (Supplementary Fig. 1a, b). To better characterize Rab30 regulation, we compared *Rab30* expression across disparate metabolic conditions. First, our analysis of *Rab30* expression in the livers of wildtype mice under different dietary states shows that *Rab30* mRNA is induced specifically in the fasting state in the mouse liver as compared to chow-fed, re-feeding after a fast, high fat diet, and ketogenic diet (Fig. 1a)[19]. We next turned to mouse models of disrupted lipid metabolism and compared *Rab30* expression in models of inhibited fatty acid oxidation with Cpt2$^{L-/-}$ mice and inhibited triglyceride hydrolysis with adipose triglyceride lipase liver-specific knockout (Atgl$^{L-/-}$) animals. While both models exhibit a fatty liver following a fast[20], only the loss of Cpt2 induced *Rab30* above control levels (Fig. 1b), with fed state *Rab30*

mRNA levels similar between Cpt2$^{L-/-}$ and littermate controls (Supplementary Fig. 1c). These data show that Rab30 expression in the fasting state is exacerbated by a loss of fatty acid oxidation.

Compared to inhibition of triglyceride hydrolysis in Atgl$^{L-/-}$, the loss of fatty acid oxidation by knockout of Cpt2 in the liver uniquely induces a Pparα transcriptional response[20]. As other groups have observed Rab30 to be among the top genes regulated by Pparα during fasting in the liver[14–17], we examined the requirement of Pparα in the regulation of Rab30 in fatty acid oxidation deficiency in the livers of Cpt2$^{L-/-}$;Pparα$^{-/-}$ double-knockout mice (Fig. 1b–d). The loss of Pparα in a whole-body knockout (Pparα$^{-/-}$) failed to induce Rab30 in the liver following a fast and, furthermore, the loss of Pparα in Cpt2$^{L-/-}$:Pparα$^{-/-}$ double knockout mice ablated the *Rab30* mRNA and protein induction by fasting. Finally, activating Pparα pharmacologically is sufficient to induce *Rab30* mRNA in primary mouse hepatocytes (Fig. 1e). Taken together, these data show that Pparα is both required and sufficient to induce Rab30 in the liver following a fast.

### Rab30 is localized to dynamic membranes from the Golgi apparatus
Rab30 has been reported to be a bona fide Golgi Rab[21–23]. Early dynamics studies in HeLa cells using photobleaching suggested that Rab30 is recruited to the Golgi from the cytosol but is not involved in vesicular or tubular exit from the Golgi[21], while more recent studies indicate that it localized to vesicles through the endolysosomal pathway, including endosomes cycling the integral membrane protein TGN38 (human ortholog of mouse Tgoln1) between the Golgi and plasma membrane[24], dynamic small intracellular vesicles marked by Tumor protein D52-like family members[25–27], and even starvation and pathogen-induced autophagosomes[28,29]. BioID and other protein-protein interaction approaches in cultured human and *Drosophila* cells have also suggested Rab30 to interact with components of the exocyst and the Golgi-associated retrograde protein (GARP) complexes, indicating a potential for its involvement in the secretory and endocytic pathways[30,31]. Furthermore, Rab30 is observed to be localized to other membranes, such as to the apical pole in the Drosophila salivary gland, where it and other Rabs likely regulate Myosin V-dependent apical protein transport also in response to phosphatidylinositol levels[28,32]. The latter observation provides an interesting prospect of Rab30 as part of the machinery that maintains efficient vectorial transport and apical-to-basolateral polarity in mammalian epithelial cells such as hepatocytes. Therefore, given we understand how Rab30 expression is regulated in the mouse liver, we sought to define Rab30 localization, dynamics, and interactome in hepatocytes.

In order to better understand the localization of Rab30 in hepatocytes, we generated stably transfected AML12 cells, a cultured hepatic cell line, expressing an mScarletI-tagged Rab30. We observe that mScarletI-Rab30 localizes with the live Golgi stain C6-NBD-Ceramide in AML12 cells without perfect overlap (Fig. 2a). We next generated liver-specific adeno-associated viral vectors (AAV8) encoding mScarletI-tagged Rab30. Freshly isolated primary mouse hepatocytes expressing AAV-mScarletI-Rab30 demonstrated similar expression as AML12 cells in which mScarletI-Rab30 appears to be closely associating to but not overlapping with the Golgi stain (Fig. 2b, c). These observations are recapitulated in vivo with immunofluorescence staining of the cis-Golgi marker GM130 in the livers of fasted mice expressing AAV-mScarletI-Rab30, and we also find mScarletI signal to be polarized further out in the cytoplasm of hepatocytes (Fig. 2d).

To understand if Rab30-positive membranes are dynamically fusing to and exiting from the Golgi, we performed live-cell super-resolution imaging of mScarletI-Rab30 in primary mouse hepatocytes (Fig. 2e, Supplementary Movies 1–3). We captured dynamic budding of Rab30 vesicles and transient interactions with the Golgi: for example, an entire membrane protraction and vesiculation event of mScarletI-

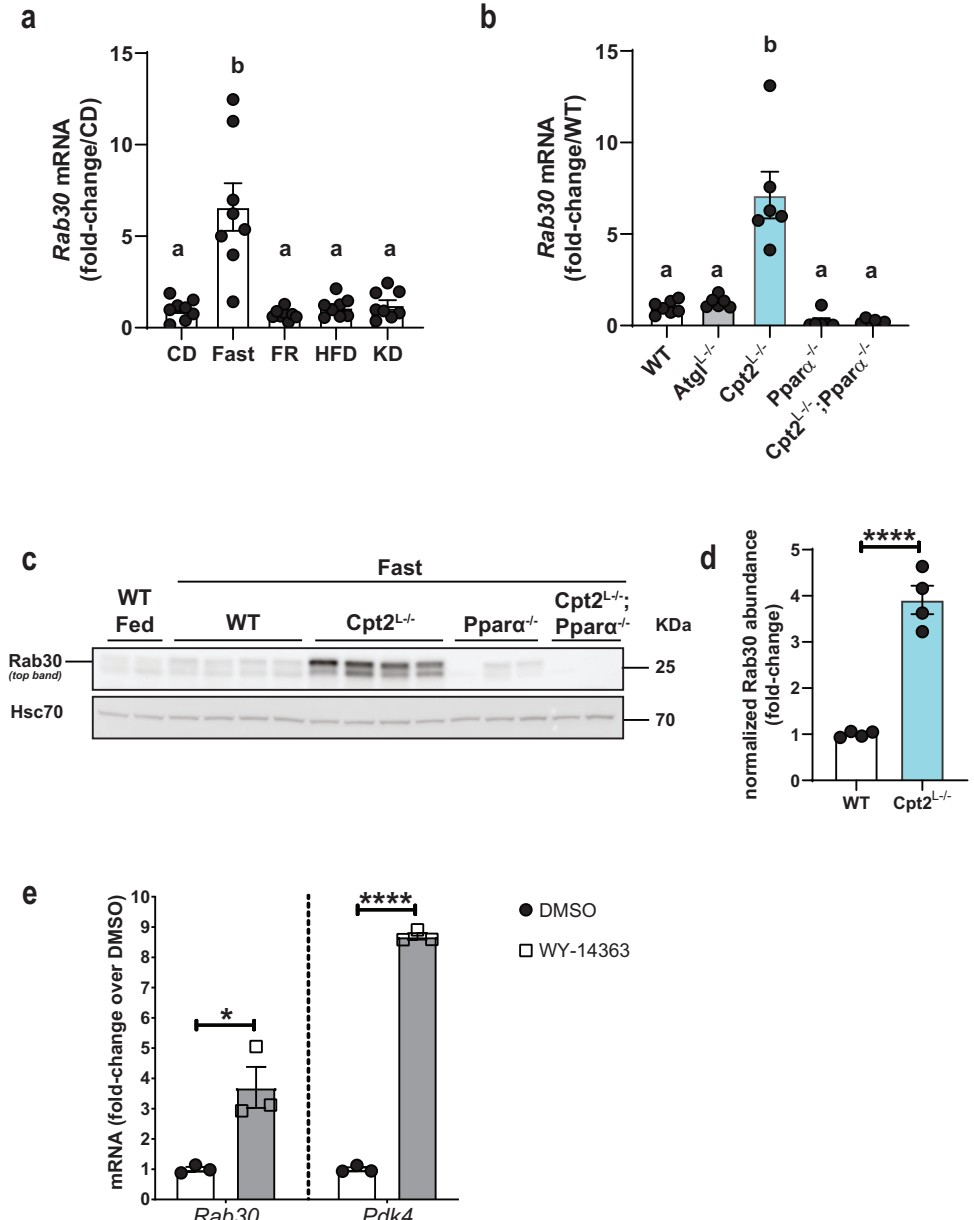

**Fig. 1 | Rab30 is induced in the mouse liver upon fasting by Pparα. a** Hepatic *Rab30* mRNA from 20-week-old male C57Bl/6J mice under different dietary states. $n = 8$/group. CD control diet, Fast CD fed mice fasted overnight, FR CD fed mice fasted for 14 h then refed for 12 h; HFD high fat (60%) diet, KD ketogenic diet. Mice were placed on the respective diets for 12 weeks[19]. Values are mean ± SEM relative to CD. Letters indicate significance groups, where different letters represent statistical significance of $p < 0.05$ following analysis by Tamhane's T2 multiple comparisons test after Welch and Brown–Forsythe ANOVA. **b** Hepatic *Rab30* mRNA in fasted male mice ($n = 7$ for WT, $n = 6$ for all other genotypes). Values are mean ± SEM relative to WT fasted. Letters indicate significance groups by Tukey's multiple comparisons test following one-way ANOVA. WT = Cpt2 floxed. **c** Representative immunoblots of Rab30 expression in fed and 24 h fasted livers of male mice with Hsc70 as an equal protein loading control. WT = Rab30;Cpt2 floxed. **d** Quantitation of Rab30 band intensity of fasted WT and Cpt2[L-/-] samples ($n = 4$ males/genotype) in **d**, normalized to Hsc70 signal and represented as fold-change over WT signal. Values are mean ± SEM. Significance was determined by two-tailed unpaired *t*-test. ****$p = 8.34 \times 10^{-5}$. **e** *Rab30* mRNA in primary mouse hepatocytes treated with DMSO vehicle control or the selective Pparα agonist WY-14643. Primary mouse hepatocytes (500,000 cells/one well of a 6-well plate) were treated with 10 μM WY-14643 or DMSO in Medium 199 supplemented with 10% FBS and 1% penicillin-streptomycin 100x solution for 16 h. Values are mean ± SEM relative to expression under control DMSO treatment. Significance was determined by two-tailed unpaired *t*-test comparing 3 wells of DMSO treated cells to 3 wells of WY-14643 treated cells for each gene. *$p = 0.0167$; ****$p = 4.08 \times 10^{-7}$. *Pdk4* is a positive control for Pparα induction after WY-14643 administration. ANOVA tables and source data for relevant panels are provided as a Source Data file.

Rab30 membranes occurs in less than 50 s. An independent live-cell super-resolution experiment in which primary mouse hepatocytes expressing mScarletI-Rab30 and stained with a live cell Golgi marker provides stronger evidence in support of Rab30-marked membranes both dynamically interacting with and also budding from the C6-NBD-ceramide stained Golgi (Supplementary Movies 2, 3).

Given our observations of Rab30 distribution in hepatocytes, we wanted to identify what proteins interact with Rab30 and their intra-cellular localization. First, we performed a yeast two-hybrid screen of a constitutively-active Rab30 mutant (Q68L, amino acids 1–198 and lacking prenylation site) against a human liver cDNA library. We identified a small number of putative direct interactors including the

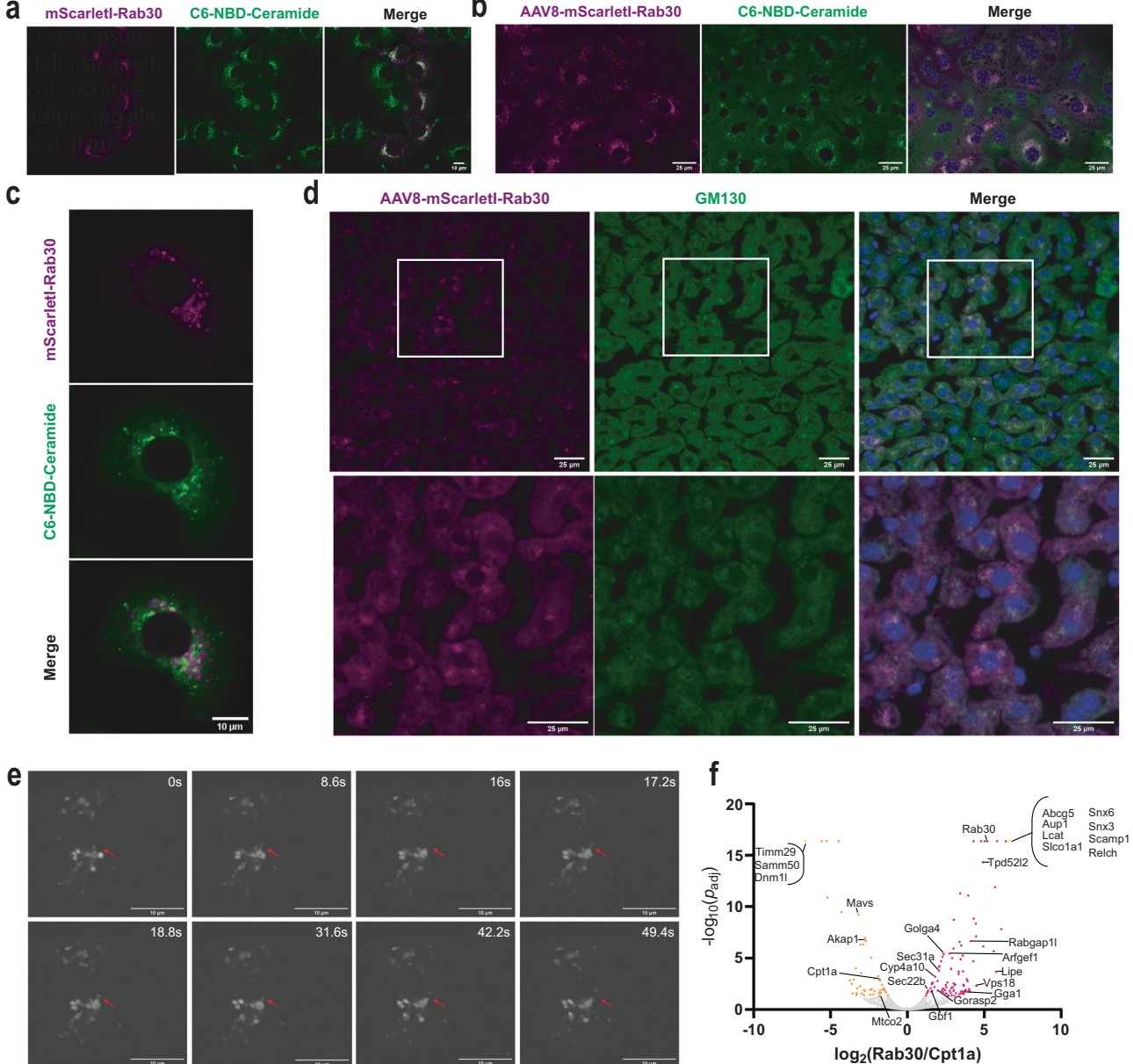

**Fig. 2 | Rab30 is localized to dynamic membranes from the Golgi. a** Confocal microscopy image of AML12 cells stably expressing HA-mScarlet-Rab30 and stained with the live-cell Golgi marker C6-NBD-Ceramide. The experiment was repeated twice with cells in replicate wells. **b** Confocal microscopy image of mouse hepatocytes isolated from animals overexpressing with AAV8-mScarletI-Rab30 and stained with the live-cell Golgi marker C6-NBD-Ceramide and Hoechst 33342 nuclear stain. Experiment was performed once in replicate wells. **c** Zoomed-in image of a representative primary mouse hepatocyte overexpressing AAV8-mScarletI-Rab30 and stained with the live-cell Golgi marker C6-NBD-Ceramide from the experiment described in **b**. **d** Immunostaining of GM130 in 24 h fasted mouse liver tissue overexpressing AAV8-mScarletI-Rab30. Blue channel in the merge is Hoechst 33342 nuclear stain. The experiment was repeated in 2-3 sections from 2 different animals. **e** Still images from time lapse of primary mouse

hepatocytes overexpressing AAV8-TBG-mScarletI-Rab30 following the progression of a putative mScarletI-Rab30-positive membrane protraction event (marked by red arrows) captured using the spinning disk confocal microscope CSU-W1 SoRa. 60x objective. The experiment was repeated twice with hepatocytes from 2 different animals. See also Supplementary Movies 1–3. **f** Volcano plot of proteins enriched in TurboID-Rab30 or -Cpt1a pulldowns. The bait proteins are highlighted in blue. Proteins detected exclusively in one pulldown are collapsed into one point marked in yellow. Significantly enriched proteins ($p_{adj} < 0.05$) are colored magenta for Rab30 and orange for Cpt1a. Statistical significance is reported as the adjusted $p$-value using the Benjamini−Hockberg correction for the false discovery rate (FDR). See Supplementary Data 2 for the list of proteins used to generate the plot. Source data are provided as a Source Data file.

Golgi coiled-coil membrane protein GOLGA5, consistent with a Golgi-specific function; somewhat unexpectedly, we detected albumin and the apolipoproteins B and A2, which we supposed would have been internal to a vesicle (Supplementary Data 1). Next, to identify the Rab30 interactome specifically in fasted state hepatocytes in vivo, we used the biotin-dependent proximity-labeling approach TurboID[33]. While we cannot differentiate between direct and indirect interactors

by this technology, understanding what proteins are in the vicinity of Rab30 provides us with insight into potential processes that Rab30 may be involved in via association. To express the biotin ligase fusion protein in vivo, we generated liver specific AAV8s encoding TurboID-Rab30 and also TurboID-Cpt1a. We used TurboID-Cpt1a, which localizes to the outer mitochondrial membrane, as a compartmental control. We expressed each vector in 4 mice, then fasted the TurboID-

Rab30 and -Cpt1a expressing mice for 21 h prior to injecting them with biotin and allowing a 3 h labeling period. At the end of the 3 h labeling period, we collected the livers, harvested total protein, and performed streptavidin pulldowns to enrich for biotinylated proteins. Western blot analysis from the livers of TurboID expressing animals and from streptavidin enrichment of biotinylated proteins shows both AAV expression and significant total biotinylation as compared to EGFP-injected animals (Supplementary Fig. 2a, b).

We used mass spectrometry to identify the proteins that bound to the streptavidin beads for each replicate. 278 proteins were significantly enriched in at least 2 replicates of TurboID-Rab30 samples, while 226 were enriched in at least 2 replicates TurboID-Cpt1a samples (Fig. 2f, Supplementary Data 2, Supplementary Data 3). The bait proteins Rab30 and Cpt1a were both significantly enriched in their respective pulldowns (Fig. 2f, Supplementary Fig. 2c). To identify the subcellular localization associated with potential interactors, the list of significantly enriched interactors for each TurboID fusion protein were submitted to the DAVID Bioinformatics Database[34,35]. As anticipated, we found that TurboID-Cpt1a interactors were significantly associated with the mitochondria, such as Samm50, Dmn1l, Timm29, Akap1, Mavs, and Mtco2 (Fig. 2f, Supplementary Fig. 2d). TurboID-Rab30 interactors, on the other hand, were more broadly associated with the secretory pathway, such as the ER, Golgi apparatus, endosome, vesicles, and cell junctions (Supplementary Fig. 2d).

Golgi-localized proteins and small GTPase effectors such as Golga4, Gorasp2, Rabgap1l, Gbf1, Arfgef1, and Gga1 were identified to be enriched in streptavidin pulldown samples for Rab30. Vesicle trafficking proteins such as Snx3, Snx6, Scamp1, and Relch were found exclusively in the streptavidin pulldowns for TurboID-Rab30 expressing animals, and Sec22b, Sec31a, Tpd52l2 (Tpd54), and Vps18 were also found to be significantly enriched. Interestingly, we also found proteins with functions in lipid and lipoprotein metabolism to be enriched as potential interactors for Rab30, such as such as the neutral sterol transporter localized to the bile canaliculus, Abcg5; the bile acid transporter localized to the basolateral membrane, Slco1a1; the hepatic secreted enzyme that esterifies the free cholesterol of lipoproteins, Lcat; a lipid droplet regulator and very low density lipoprotein assembly factor, Aup1; the triglyceride lipase, Lipe (Hsl); and the Pparα target gene Cyp4a10. Additionally, the Golgi-localized proteins Golga4 and Gorasp2 have been implicated in the regulation of lipoprotein metabolism previously[36,37]. Correspondingly, "lipid metabolic process" and "fatty acid metabolic process" are Gene Ontology terms associated with enriched interactors with Rab30 (Supplementary Fig. 2e). Together, these imaging and biochemical interaction data demonstrate that Rab30 is a dynamic interactor with the Golgi apparatus and also with proteins throughout the secretory pathway, likely localizing to cytoplasmic and post-Golgi vesicles in mouse hepatocytes. Furthermore, the interaction between Rab30 and known regulators of lipid metabolism supports the hypothesis for a role of Rab30 in contributing to hepatic lipid homeostasis. Further studies will be required to validate the location of the interactions and understand the functional role in hepatocyte biology. We now focus on the physiological implications of Rab30 on hepatic lipid metabolism.

## Generation of Rab30 knockout mice

Given our observations that Rab30 (1) is a target of Pparα, a master transcriptional regulator of lipid catabolism and hepatocyte energy balance in the fasted state, (2) is dramatically upregulated in mice with deficient hepatic fatty acid oxidation, and (3) dynamically associates with the Golgi and proteins along the secretory pathway and regulators of lipid metabolism in live mouse hepatocytes, we hypothesized that loss of Rab30 in vivo might affect the secretion and/or turnover of lipids and proteins from the liver. Therefore, to determine the physiological role of Rab30 in response to fasting in the mouse liver, we generated Rab30 knockout (Rab30KO) and conditional

knockout mouse (Rab30 ff) lines by CRISPR-Cas9 (Fig. 3a). Rab30KO mice were viable and fertile with an absence of *Rab30* mRNA and protein (Fig. 3b, c). Mice containing the floxed *Rab30* alleles were mated to mice expressing the albumin-Cre transgene to specifically knockout Rab30 in hepatocytes (Rab30[L−/−]) (Fig. 3d, e). Because we see a potentiated Pparα response in the livers of Cpt2[L−/−] fasted mice and because Rab30 expression depends on the activity of Pparα, we reasoned that we could use the genetic background of the Cpt2[L−/−] animals to amplify the effect of the loss of Rab30 in the liver. Therefore, Rab30 floxed mice were also mated to Cpt2 floxed animals carrying the albumin-Cre transgene to create liver-specific Rab30;Cpt2 double knockout (DKO) mice to determine the role of Rab30 induction in Cpt2[L−/−] mice (Fig. 3d, e). The loss of Rab30 in the germline, the liver, or in Cpt2[L−/−] liver had minimal effects on fed and fasting bodyweight, apart from Rab30KO males being larger on average than control males by 1.77 g in the fasted state (Supplementary Fig. 3a, c). Loss of Rab30 did not affect fed or fasting blood glucose measurements in males (Supplementary Fig. 3b); however, female blood glucose was increased in fasted Rab30KO female mice compared to littermates but decreased in DKO females compared to their controls (Supplementary Fig. 3d). While these data illuminate a potential sex difference in the response to fasting mediated by Rab30, we did not pursue further investigation of these discrepancies here because they are not related our main hypothesis regarding a role for Rab30 in the regulation of Pparα-dependent lipid and protein trafficking.

The influence of Rab30 on Golgi-ultrastructure remains a point of contention throughout the literature. In cultured cells, siRNA knockdown of Rab30 revealed a Golgi-dispersion phenotype, but knockout of Rab30 by CRISPR/Cas9 in transformed cells showed no fragmentation of the Golgi[21,24,38]. To understand the contribution of Rab30 on Golgi structure in the mouse in vivo, we visualized the Golgi by transmission electron microscopy in livers of control, Rab30KO, and Rab30; Cpt2[L−/−] DKO mice and the hippocampus of control and Rab30KO mice (Fig. 3f, g). The Golgi apparatus appeared similar in both tissues between Rab30 knockout and wildtype mice. In conjunction with the data presented in Fig. 2, these data show that Rab30 associates with the Golgi but is not required for its ultrastructure in vivo.

Pparα regulates autophagosome formation in the mouse liver in response to fasting[3–5]. Given that the Golgi is also an important source for autophagic membrane lipids and proteins[39], we asked if Rab30 was downstream of Pparα-directed autophagy. Fasting mRNA for autophagy related proteins, including *Beclin-1* and *LC3B*, proteins responsible for the initiation of autophagosome formation, were found to be slightly increased (i.e., fold-change roughly <1.5) in both the Cpt2[L−/−] and DKO livers to similar extents, except for *Lamp2*, which was decreased in DKO livers (Supplementary Fig. 4). We next performed immunoblots against Beclin-1 and LC3A/B in wildtype fed livers and the fasted livers of wildtype, Rab30[L−/−], Cpt2[L−/−], Rab30;Cpt2 DKO, and Pparα[−/−] animals (Fig. 3h). Beclin-1 and lipidated LC3 protein levels are not affected in fasted Rab30 knockout livers as compared to fed wildtype livers or fasted Pparα knockout livers, which exhibit decreased levels of both proteins. Taken together, our data does not support a role for Rab30 in Pparα-mediated autophagosome formation. However, as previously mentioned, Rab30 has been found to co-localize to autophagic membranes in HeLa cells[28,29], and so it is possible that Rab30 could localize to the autophagosome following initiation and elongation or that the isolation membrane is scavenged from other organelles with the loss of Rab30. Another well-known role for the Golgi in regulating fatty acid metabolism is by affecting the processing of sterol regulatory element-binding protein 1 (Srebp1). Therefore, we determined the regulation of Srebp1 target genes in wildtype, Rab30[L−/−], Cpt2[L−/−], Rab30;Cpt2 DKO livers in the fed and fasted states (Fig. 3i). As Srebp1 target gene expression is largely unchanged with the loss of Rab30, we found no evidence to support a role of Rab30 in mediating Srebp1 processing.

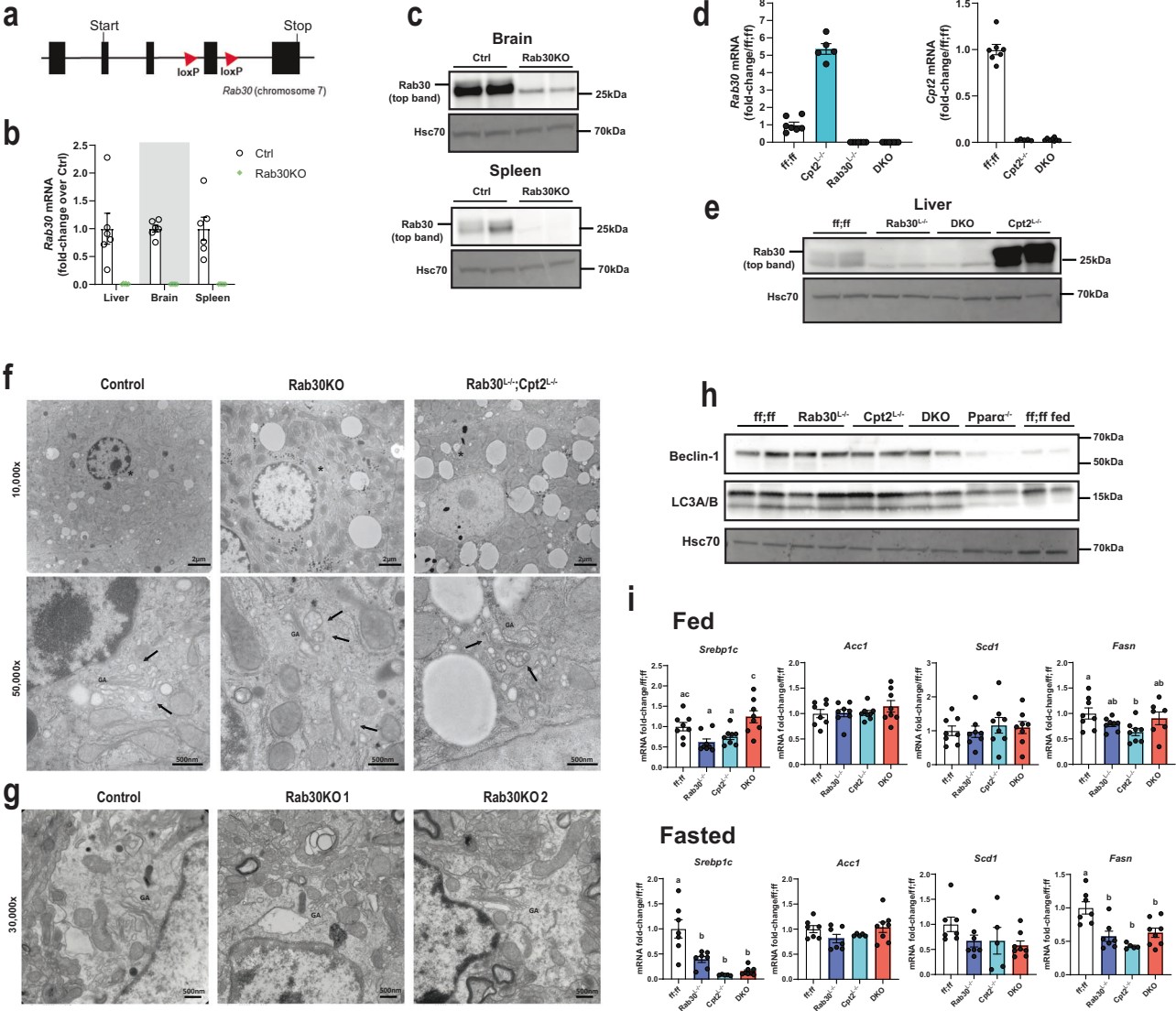

**Fig. 3 | Characterization of Rab30 knockout mice. a** Gene targeting strategy for the *Rab30* gene, with *loxP* site insertion indicated by triangles. **b** Fasting *Rab30* mRNA in the livers, brains, and spleens of whole-body Rab30 knockout male mice (Rab30KO) and littermates (Ctrl) ($n = 6$/genotype). Values are mean ± SEM relative to Ctrl for a given tissue. **c** Rab30 immunoblot in the brain and spleen of Rab30KO and control males. Hsc70 is an equal protein loading control. **d** Fasting hepatic qRT-PCR for *Rab30* (left) and *Cpt2* (right) mRNAs in male livers. Values are mean ± SEM relative to Rab30;Cpt2 floxed (ff;ff) animals. DKO = Rab30$^{L-/-}$;Cpt2$^{L-/-}$. $n = 5$ for Cpt2$^{L-/-}$, $n = 7$ for ff;ff, and $n = 8$ for DKO. **e** Rab30 immunoblot in the livers of Rab30$^{L-/-}$, Cpt2$^{L-/-}$; DKO, and control (ff;ff) males. **f** Representative transmission electron micrographs of liver cells from 5–6-week-old 24 h fasted male knockouts and control males. Asterisks (*) in 10,000x denote region of 50,000x acquisition. GA, Golgi apparatus; arrows, vesicles. The experiment was performed in a total of 5 controls and 3 of each knockouts. **g** Representative transmission electron micrographs of hippocampi from 7-8-week-old control and 2 Rab30KO females under basal conditions taken at 30,000x. GA Golgi apparatus. The experiment was performed in a total of 3 females per genotype. **h** Western blot for autophagosome initiation markers in the livers of control fed male mice and fasted control (ff;ff), Rab30$^{L-/-}$, Cpt2$^{L-/-}$, Rab30;Cpt2 DKO, and Pparα$^{-/-}$ male mice. Hsc70 is a protein loading control. **i** qRT-PCR of *Srebp1c, Acc1, Scd1,* and *Fasn* in fed and fasted livers of control (ff;ff), Rab30$^{L-/-}$, Cpt2$^{L-/-}$, and DKO males. $n = 8$/genotype for fed. $n = 5$ for Cpt2$^{L-/-}$, $n = 7$ for ff;ff and Rab30$^{L-/-}$, and $n = 8$ for DKO for fast. Values are mean ± SEM. Letters indicate significance groups by Tukey's multiple comparisons test following one-way ANOVA for each individual gene. ANOVA tables and source data for relevant panels are provided as a Source Data file.

## Loss of Rab30 results in retention of secreted proteins

Next, we performed RNA-seq and proteomics on 24 h fasted livers from Rab30 knockouts, wildtype, and Cpt2$^{-/-}$ animals for two main reasons. First, we were concerned that loss of Rab30 might influence the expression of other Rab proteins. Second, we predicted that if Rab30 was involved in intracellular membrane trafficking from the Golgi, then we would see a buildup of secreted proteins in the livers of Rab30 knockout mice.

Global analysis of the liver transcriptome by principle component analysis of normalized transcript levels revealed that the Rab30$^{L-/-}$ and Rab30KO clustered together with Rab30;Cpt2 floxed (ff;ff) and Rab30KO littermate controls (Rab30 Ctrl), while Cpt2$^{-/-}$ and Rab30;Cpt2 DKO animals clustered together but apart from the single knockouts (Fig. 4a). These data are consistent with liver proteomic data that largely shows gene specific clustering (Fig. 4b). As we expected, these data suggest that loss of Rab30 does not drastically affect the fasted liver transcriptome.

We directly investigated the transcriptomics and proteomics data set for differentially expressed Rab family members (Fig. 4c, d, Supplementary Fig. 5a). There were no significantly changed Rabs in the transcriptomics dataset when comparing Rab30KO livers to littermate control livers. When comparing Rab30$^{L-/-}$ to control (ff;ff) livers or

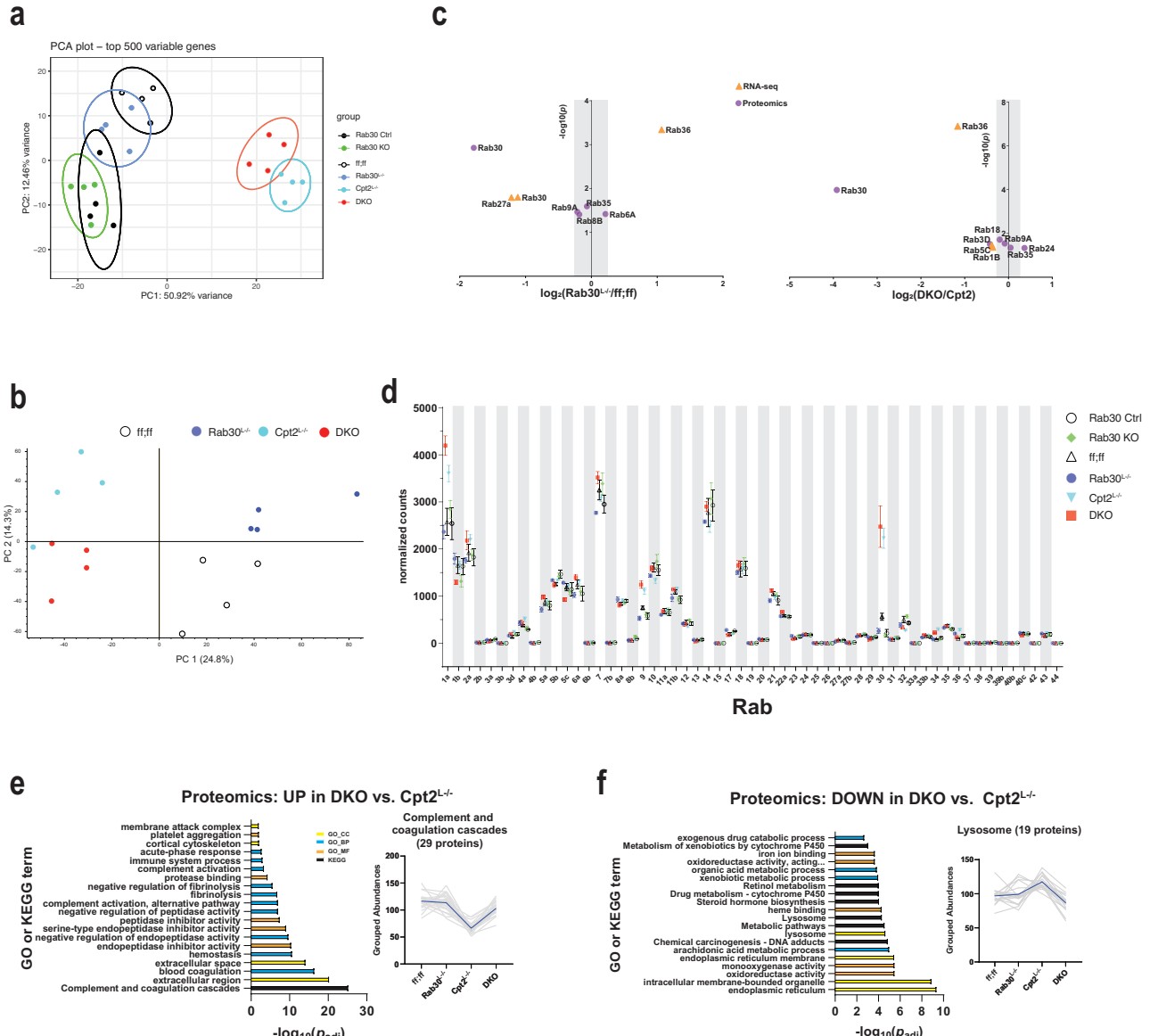

**Fig. 4 | RNA-seq and proteomics analysis of 24 h fasted livers. a** Principal component analysis of RNA-seq on 24 h fasted livers from Rab30;Cpt2 floxed (ff;ff), Rab30KO littermate controls (Ctrl), Rab30$^{L-/-}$, Cpt2$^{L-/-}$, DKO, and Rab30KO male mice (n = 4/genotype). **b** Principal component analysis of proteomics on 24 h fasted livers from Rab30;Cpt2 floxed (ff;ff), Rab30$^{L-/-}$ (Rab30), Cpt2$^{L-/-}$, and DKO mice (n = 4/genotype). **c** Volcano plot of significantly differentially expressed Rabs between Rab30$^{L-/-}$ and control (ff;ff) or DKO and Cpt2$^{L-/-}$ 24 h fasted livers in the RNA-seq (orange triangle) and proteomics (purple circle) datasets. Points outside of the gray bar represents a fold-change of at least 1.2. **d** Normalized read counts of Rab family members in RNA-seq of 24 h fasted livers. Values are the average of 4 samples/genotype ± SEM. Gene ontology and KEGG pathway analysis of

proteomics comparing up- (**e**) and down- (**f**) regulated pathways in the DKO vs. Cpt2$^{L-/-}$. Proteins with fold-change of at least 1.2 and p < 0.05 were submitted for pathway analysis to the DAVID functional annotation tool. Significantly enriched pathway terms are ranked against the Benjamini-adjusted p-value generated by the DAVID functional annotation tool. Top 20 terms by $p_{adj}$ are presented. Colors represent pathway class as denoted in the legend. Normalized and scaled abundances vs. genotype plots depict the traces of all identified proteins within the given pathway, with the black line indicating the average grouped abundance of the proteins. For (**e**) mmu04610:Complement and coagulation cascades. For **f** GO:0005764-lysosome. p-values for pathway analysis and source data for relevant panels are provided in the Source Data file.

DKO to Cpt2$^{L-/-}$ livers, we find a few changes in Rab mRNA and protein abundances; most were not greater than 1.2 fold-change (Fig. 4c, d, Supplementary Fig. 5a). In summary, loss of Rab30 does not appear to significantly alter the expression of other Rab family members in the liver. These data suggest that, at least at the expression level, the loss of Rab30 does not result in compensatory changes in other Rab family members; however, we do not know if loss of Rab30 affects their functions independently of expression level.

We then used the DAVID Bioinformatics Database[34,35] to perform pathway analysis of up- and down-regulated proteins from our proteomic dataset. When comparing Rab30$^{L-/-}$ mice to control mice, we

found no significantly enriched pathway terms upregulated in the liver knockouts. However, when comparing DKO mice to Cpt2$^{L-/-}$ mice, we find that there are pathway terms associated with hepatic secreted proteins upregulated in the fasted DKO livers (Fig. 4e). The GO cellular component term "GO:0005576-extracellular region" was the second-most enriched term by p-value and was comprised of 82 proteins. The KEGG term "mmu04610:Complement and coagulation cascades" was the most enriched by p-value and contained 29 proteins. This pathway is not significantly enriched on the transcriptomic level when comparing Cpt2$^{L-/-}$ to DKO (Supplementary Fig. 5b), indicating that the difference is not transcriptional in nature and possibly points to a

defect in protein secretion and/or turnover. Furthermore, this term was also strongly downregulated on transcriptional levels in both the Cpt2[L−/−] and DKO fasted livers when compared to control animals; while this downregulation is upheld on the proteomic level for Cpt2[L−/−] mice, the loss of Rab30 in the DKO brings the protein abundances to about wildtype levels (Fig. 4e, trace of grouped abundances, and Supplementary Fig. 5c).

In terms of downregulated proteins, we find that proteins with pathway terms associated with the endoplasmic reticulum, intracellular membrane bounded organelle, and lysosome were significantly enriched in DKO animals compared to Cpt2[L−/−] (Fig. 4f). Specifically, the abundances of lysosomal resident or lysosomal-interacting proteins with the GO term "GO:0005764-lysosome" such as Lamp2, Syt11, Lipa, and Ctsf are decreased in DKO livers. Taken together, these data possibly indicate a disruption of trafficking from the Golgi with the loss of Rab30. We rationalize that we only observe this phenotype in the DKO livers because the demand for lipid and protein recycling for energetic substrates is exacerbated in hepatocytes that cannot generate ATP from lipids or rid the liver of excess fat through of β-oxidation, and perhaps there is increased demand of lysosomal degradation of proteins and lipid stores in the absence of Rab30-dependent membrane trafficking. In livers that are able to oxidize fats, the loss of Rab30 may be minimized due to increased metabolism or compensated by other mechanisms that maintain protein flux through the Golgi.

## Rab30 influences circulating triglyceride and cholesterol

We next questioned if the defect in Golgi and endolysosomal trafficking of proteins could also relate to a role for Rab30 the mobilization of lipids in the liver in a Pparα-dependent manner. Therefore, we investigated if loss of Rab30 impacted lipid homeostasis in the liver and the serum of fasted mice. At the tissue level, the Cpt2[L−/−] and DKO livers are quantifiably larger and visibly paler than the control and Rab30[L−/−] livers (Fig. 5a, b). The basis for the dramatic increase in liver:body weight ratios after a fast in the Cpt2[L−/−] and DKO mice is predominantly due to fasting-induced hepatic lipid accumulation, as evidenced by the following rationale. First, there are no differences in liver:body weight ratios between knockouts and their controls in the fed state male mice (Supplementary Fig. 3e), and is consistent with previous observations described by our lab when initially characterizing Cpt2[L−/−] mice[18]. Next, examination of H&E stained sections of fed livers from control, Rab30[L−/−], Cpt2[L−/−], and DKO mice revealed no differences between genotypes (Supplementary Fig. 3f), while the fasting state histology showed that the Cpt2[L−/−] and DKO sections were comprised of cells that appeared to be swollen and lipid laden compared to the control and Rab30[L−/−] sections (Fig. 5c). BODIPY (Fig. 5c) and Oil Red O (Supplementary Fig. 3g) staining of fasted liver sections further indicate hepatic lipid accumulation in the DKO and Cpt2[L−/−] mice. The fasted state BODIPY 493/503 stained liver sections of the Rab30[L−/−] and control livers look overtly similar, while, as expected, both Cpt2[L−/−] and DKO livers contained cells bloated with lipids (Fig. 5c). Furthermore, as observed in BODIPY 493/503 staining of lipid droplets in fasted liver tissue (Fig. 5c), fasting induced significantly more lipid droplet accumulation in the liver with the loss of hepatic zonation in the periportal region in the Cpt2[L−/−] and DKO mice as compared to control or Rab30[L−/−] mice due to the imbalance of the inability to oxidize fatty acids with loss of Cpt2 and the influx of lipids. Oil Red O staining of the liver sections show similar results to the BODIPY stained liver sections (Supplementary Fig. 3g). Finally, while there is no difference in liver triglycerides between Rab30KO and Rab30[L−/−] and their respective controls, both the Cpt2[L−/−] and DKO livers exhibit increased hepatic triglycerides (Fig. 5d). Overall, as the hepatic triglyceride content is driven by Cpt2[L−/−] and not compounded by the loss of Rab30, we conclude that loss of Rab30 does not affect total hepatic triglyceride stores.

While Rab30 is clearly enriched near the Golgi in both cultured hepatocytes and wildtype fasted livers (Fig. 2a−d) and despite the fact that hepatic triglyceride content is not altered with the loss of Rab30, we questioned if Rab30 would be driven to localize with lipid droplets in livers lacking Cpt2 in a Pparα-dependent manner. We therefore expressed mScarletI-Rab30 by AAV8 in wildtype and DKO mice and stained their livers after a 24 h fast with BODIPY 493/503 to mark lipid droplets (Fig. 5e). As previously observed in Fig. 2d, Rab30 signal is observed to be polarized from the center of the cell throughout the cytoplasm in control fasted livers and does not co-localize with the modest accumulation of lipid droplets. In the lipid-laden DKO livers that exhibit potentiated Pparα transcriptional activity, Rab30 signal does not co-localize with lipid droplets as we initially wondered, but instead appears to be enriched at the cell periphery. This plasma-membrane localization is consistent with our TurboID-Rab30 data (Fig. 2), as we identified plasma membrane receptors and transporters to be interactors with Rab30 in the mouse liver. Interestingly, when we expressed mScarletI-Rab30 in Atgl[L−/−] livers, which also present a fasting-induced fatty liver but not Pparα induction, we do not find Rab30 to be enriched at the cell periphery (Fig. 5f).

While further experimentation is required to elucidate the mechanism behind the enrichment of Rab30 at the cell periphery only in mice lacking the ability to perform mitochondrial β-oxidation but not triglyceride hydrolysis, these results could imply an increased demand for lipid excursion in a Pparα-dependent manner in fasted livers that is exacerbated in the absence of β-oxidation and that Rab30 acts as part of this lipid disposal pathway. In fact, we have previously shown that Cpt2[L−/−] mice but not Atgl[L−/−] exhibit fasting-induced serum dyslipidemia[20]. We therefore asked if loss of Rab30 has an effect on circulating lipids. We quantified triglycerides, non-esterified fatty acids, cholesterol, and β-hydroxybutyrate in the serum from Rab30KO, Rab30[L−/−], Cpt2[L−/−], and DKO male mice compared to their controls. Rab30KO animals show a fasting-induced decrease serum triglycerides compared to littermates (Ctrl) (Fig. 6a). However, when comparing the Rab30[L−/−] to their controls (ff), there are no significant differences in the serum metabolites measured (Fig. 6b), indicating that the loss of Rab30 alone in the liver is not sufficient to influence fasting serum triglyceride levels, and that the decrease observed in the whole-body knockout is due to a cumulative effect of several contributing tissues. We next quantified these lipid species in Rab30;Cpt2 floxed (ff;ff), Cpt2[L−/−], and DKO serum (Fig. 6c). Fasting ketone bodies were significantly decreased in the DKO and Cpt2[L−/−] as expected due to the suppression of ketogenesis in mice lacking hepatic Cpt2. As previously published, Cpt2[L−/−] mice have increased serum triglycerides and cholesterol levels[18]. Interestingly, the loss of Rab30 on the Cpt2[L−/−] background reverses the increased triglycerides and cholesterol levels back to control. Additionally, the DKO animals have increased NEFA compared to control and Cpt2[L−/−]. These differences are not observed in the fed state. By using the genetic background of Cpt2[L−/−] mice that exhibit a high induction of Rab30 upon fasting, we have discovered that hepatic Rab30 expression contributes to serum lipid levels during fasting.

## Loss of Rab30 impacts circulating ApoA4 abundance

As Rab30 does not localize to lipid droplets, it is not likely that Rab30 is influencing serum triglyceride levels through turnover of lipid droplets by direct interaction. Additionally, our proteomics data reveals that loss of Rab30 causes a retention of secreted proteins in the liver, indicating a potential transport defect. We therefore analyzed the serum proteome by separating serum-derived proteins on an SDS-PAGE gel and staining for total protein by Coomassie to see if we could identify secreted regulators of lipid metabolism influenced by loss of Rab30 (Fig. 7a). Overall, the banding pattern amongst genotypes appeared to be similar at the sensitivity afforded by this assay. However, we were intrigued to find that one ~45 kDa protein in particular

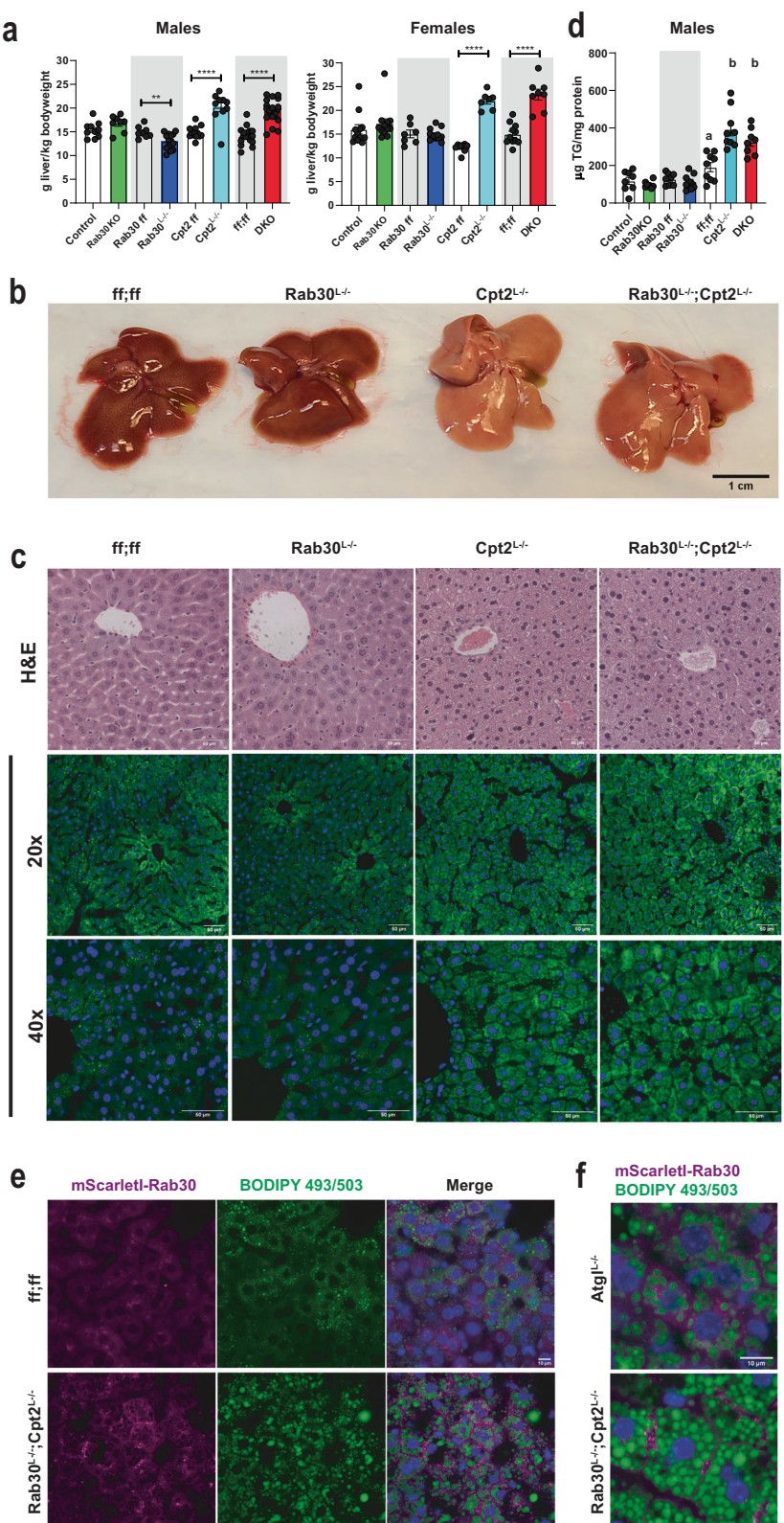

was highly abundant in the Cpt2$^{L-/-}$ serum and appeared to be suppressed in the DKO serum (band 'A'), as well as further decreased in the control and Rab30$^{L-/-}$ animals. We excised this band, as well as the ~25 kDa protein labeled 'B', which appears consistent amongst the genotypes. We identified these bands by mass spectrometry as ApoA4 (band 'A') and ApoA1 (band 'B'). We then visualized ApoA4 levels in the serum of fasted control, Rab30$^{L-/-}$, Cpt2$^{L-/-}$, and DKO males and

females by western blot (Fig. 7b, c, Supplementary Fig. 6a–d). The loss of Rab30 in the DKO suppresses the increase in the ApoA4 levels observed in the Cpt2$^{L-/-}$ serum.

We investigated the expression of ApoA4 and other apolipoproteins in the livers of fasted mice. We performed qRT-PCR from fasted male livers for *ApoE*, *ApoB*, and a gene cluster of apolipoproteins conserved in mice and humans involving *ApoA1, ApoC3, ApoA4,* and

**Fig. 5 | Loss of Rab30 does not influence hepatic triglyceride content. a** Wet liver weights (large left lobe) normalized to body weight of fasted male and female mice. Data are represented as average ±SEM. Asterisks denote significance between knockouts and their littermate controls (Control, Rab30 ff, Cpt2 ff, and ff;ff for Rab30KO, Rab30$^{L-/-}$, Cpt2$^{L-/-}$, and DKO, respectively) by two-tailed $t$-test: *$p < 0.05$; **$p < 0.01$; ***$p < 0.001$, ****$p < 0.0001$. **b** 24 h fasted livers of control, Rab30$^{L-/-}$, Cpt2$^{L-/-}$, and Rab30;Cpt2 DKO female mice. **c** 24 h fasted histology of H&E and BODIPY 493/503 stained livers from control, Rab30KO, Cpt2$^{L-/-}$, and Rab30;Cpt2 DKO male mice. Scale bar represents 50 μm. Blue in the BODIPY 493/503 images is Hoechst 33342 nuclear stain. Both experiments were performed in 2 mice per genotype. **d** Hepatic triglyceride levels in 24 h fasted male mice livers. Controls are Ctrl ($n = 8$) for Rab30KO ($n = 8$), ff ($n = 8$) for Rab30$^{L-/-}$ ($n = 9$), and ff;ff ($n = 9$) for Cpt2$^{L-/-}$ ($n = 9$) and DKO ($n = 8$) controls. Data are represented at average ± SEM.

Significance was determined by two-tailed unpaired $t$-test for Ctrl vs Rab30KO ($p = 0.32$) and ff vs Rab30$^{L-/-}$ ($p = 0.42$). Significant differences between ff;ff, Cpt2$^{L-/-}$, and DKO were determined by one-way ANOVA; letters indicate significance groups after Tukey's multiple comparisons test. **e** BODIPY 493/503 stained 24 h fasted livers from Rab30;Cpt2 floxed (ff;ff) and DKO females expressing mScarletI-Rab30 in hepatocytes by adenoassociated virus. Blue in merge is Hoechst 33342 nuclear stain. Scale bar represents 10 μm. Experiments were performed in 2 mice per genotype. **f** Comparison of mScarletI-Rab30 localization in 24 h fasted Atgl$^{L-/-}$ and DKO female livers stained with BODIPY 493/503. Experiment was performed in 1 Atgl$^{L-/-}$ female and 2 DKO females. If not reported in the legend, all n, $p$-values, ANOVA tables, and source data for relevant panels are provided in the Source Data file.

*ApoA5* (Fig. 7d, Supplementary Fig. 6e). *ApoB, ApoA1, ApoC3,* and *ApoA5* mRNA varied little by genotype in males, while *ApoE* was suppressed ~40% in Cpt2$^{L-/-}$ and DKO livers. However, *ApoA4* was the most highly regulated of the surveyed apolipoproteins and was significantly induced in the livers of both sexes of Cpt2$^{L-/-}$ and DKO animals (Fig. 7c, d). In agreement with the qRT-PCR data, transcriptomics data, and proteomics data, we observe the upregulation of hepatic ApoA4 in the Cpt2$^{L-/-}$ and DKO fasted male livers by western blot (Supplementary Fig. 6f, g). While ApoA4 is synthesized by the intestine and liver, hepatic expression of ApoA4 is hypothesized to decrease lipid burden of the liver by promoting excursion during fasting[40,41]. Therefore, these data indicate that ApoA4 could be playing a significant role in lipid turnover in the hepatocyte under the pathophysiological conditions of the Cpt2$^{L-/-}$ livers and its efficient trafficking requires Rab30 induction.

## Discussion

Hepatocytes are among the most suitable cell types in which to study the intersection between metabolism and cell biology due to their requirements for a flexible metabolic capacity and efficient intracellular trafficking system. Here, we sought to define the role of Rab30, an enigmatic mediator of vesicular transport in the liver. Rab30 is highly induced in the liver of Cpt2$^{L-/-}$ mice that cannot perform hepatic β-oxidation and therefore exhibit fasting-induced hepatic steatosis, serum dyslipidemia, and increased Pparα transcriptional activity[18]. We hypothesized that Rab30 could be a transcriptional target of Pparα and acting in processes mediating lipid and/or protein recycling to salvage energetic substrates during a fast. Investigation of *Rab30* expression in wildtype mice under disparate dietary interventions revealed that *Rab30* is specifically a fasting-induced gene in the liver, and that Pparα expression is necessary and sufficient for its expression in fasted livers. Our results are consistent with other reports showing that Rab30 is among the top genes in the liver induced during fasting in a Pparα-dependent manner[14–17,42]. Interestingly, recent analysis of Pparα function in intestinal lipid metabolism reveals that *Rab30* is upregulated after high-fat diet in the mouse intestine and that this induction also requires Pparα[43], possibly indicating a conserved function in lipid handling the liver and intestine.

Rab30 had previously been suggested to be a Golgi resident protein required for its structure[21]. Similarly to a CRISPR knockout cell line[38], we do not find altered Golgi structure in Rab30 knockout mice, and, while Rab30 associates with the Golgi, its localization does not completely overlap with Golgi markers in cell culture or hepatocytes in vivo. We observe highly dynamic Rab30-postive puncta radiating from the Golgi by live-cell super-resolution imaging. Our yeast two-hybrid and in vivo proximity labeling data also indicates that Rab30 interacts with proteins along the secretory pathway in the liver. We were able to recapitulate some previously established interactions. Previous interaction studies in HeLa cells and *Drosophila* identified human orthologs of the effectors such as USO1, OCRL, GOLGA4, GOLGA5, MICAL3, and RELCH[30,31,44], all of which were detected by one or both of our interaction methods (Fig. 2f, Supplementary Data 1 and 2). Rab30 has also

previously been found to colocalize to vesicles in cultured cells that are marked by Tpd54 (or Tpd52l2), a protein that is also significantly enriched as Rab30 interactor from our in vivo proximity labeling dataset[25–27]. Furthermore, we identify interactions between Rab30 and proteins with functions in hepatic lipid metabolism, such as Slco1a1, Abcg5, Lcat, Aup1, Lipe, and Cyp4a10. Further investigation is required to define the subcellular locations and functional consequences of these interactions in lipid metabolism in the mouse liver.

We examined the requirement of Rab30 during fasting using whole-body, liver-specific, and Rab30$^{L-/-}$;Cpt2$^{L-/-}$ double knockout mice. Because Rab30 expression is potentiated in the fasted livers of Cpt2$^{L-/-}$ mice, we hypothesized that the role of Rab30 might also be augmented in Cpt2$^{L-/-}$ livers. Indeed, while whole body knockouts of Rab30 did exhibit decreased serum triglyceride levels following a fast, Rab30 single liver knockouts did not overtly appear to have an influence on fasting triglyceride levels or impact steady-state secreted protein levels. However, by using the DKO model to amplify the effect of loss of Rab30 in the liver specifically, our data shows a retention of secreted proteins in double knockout livers, indicating a disruption of hepatocyte Golgi trafficking, and that circulating triglycerides and cholesterol levels are altered with the loss of Rab30. Cpt2$^{L-/-}$ mice exhibit hypertriglyceridemia and hypercholesterolemia after a 24 h fast, likely due to increased excursion of triglycerides from the liver in the absence of hepatic β-oxidation to dispose of the fasting-induced influx of lipids from adipose tissue lipolysis. Loss of Rab30 in the Cpt2$^{L-/-}$ liver rescues the dyslipidemia and serum cholesterol levels, without altering hepatic triglyceride content. We also observe an increase in serum NEFA levels in the DKO mice. The decrease in serum triglyceride levels corresponds to a decrease in circulating ApoA4 abundance.

ApoA4 was the only significantly induced apolipoprotein at the level of mRNA and protein in mice lacking Cpt2 (Fig. 7d, Supplementary Fig. 1b, Supplementary Fig. 6f, g). The function of ApoA4 in lipoprotein metabolism, especially in the liver, is itself unclear. ApoA4 is synthesized in the liver and intestine[45,46]. The hypolipidemic phenotype originally observed in ApoA4 knockout mice is likely due to a simultaneous disruption in ApoC3. A more discrete TALEN-mediated ApoA4 knockout in Sprague-Dawley rats with normal ApoC3 expression demonstrated a fasting specific hepatic steatosis[47,48]. Therefore, ApoA4 is likely important for recycling of fatty acid between liver and peripheral tissues in the fasted state. The loss of ApoA4 in models of hepatic steatosis decreases VLDL secretion from the liver, and, conversely, overexpression of ApoA4 was found to increase VLDL secretion rate and particle size[40]. These data suggest that ApoA4 promotes lipid excursion from the liver in the face of steatosis and requires Rab30 for its efficient trafficking, at least during fasting.

Based on our current data, we propose a physiological role for Rab30 during fasting in the repackaging of adipose-derived lipids in the liver and their redistribution as lipoproteins to supply other organs with an energy source during fasting. While the assembly of lipoproteins is initiated at the ER, particles are further modified in the Golgi, making it an important organelle for maintaining cholesterol

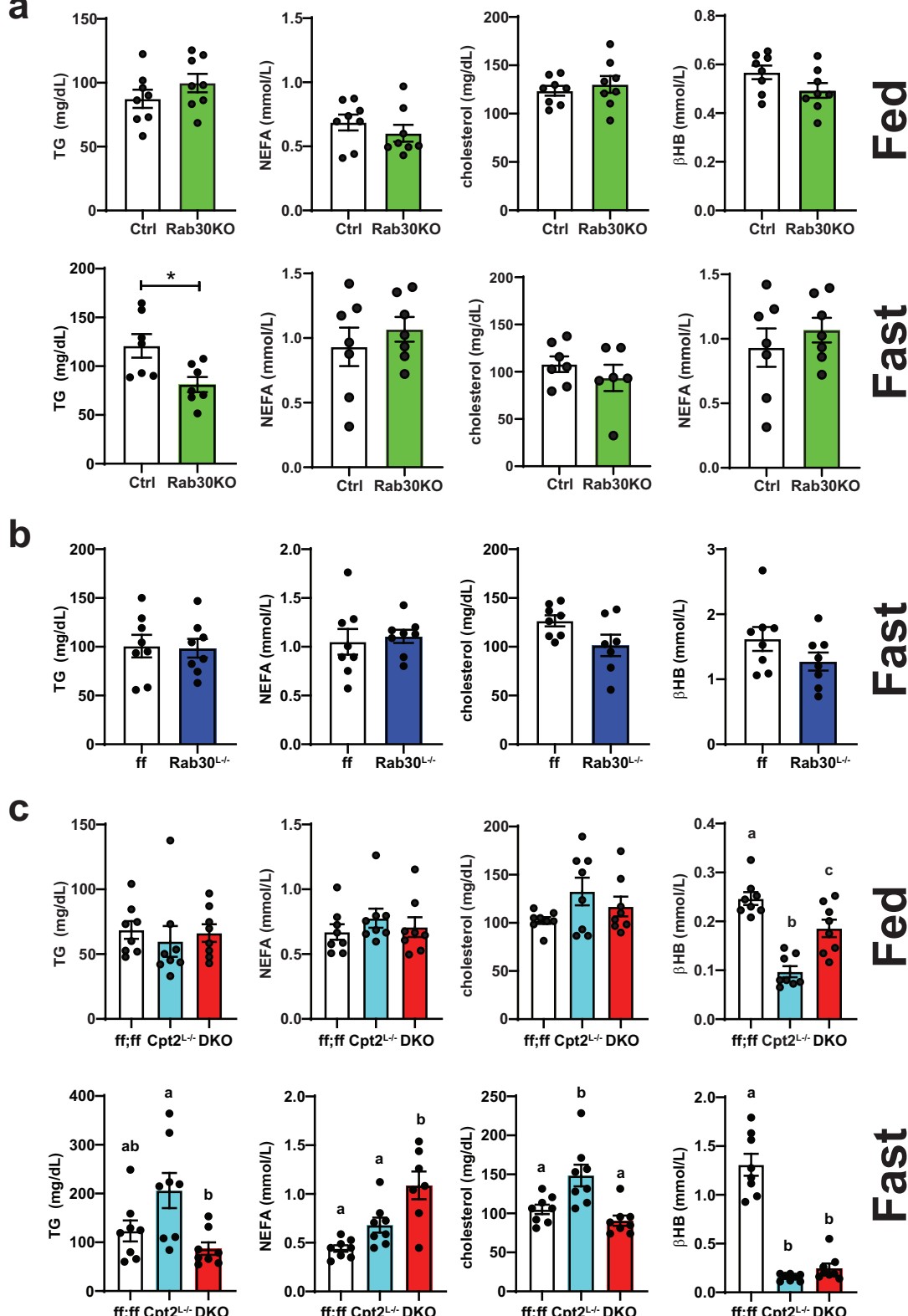

**Fig. 6 | Loss of Rab30 influences fasting circulating triglyceride and cholesterol and suppresses dyslipidemia in fatty acid oxidation deficiency. a** Fed and fasting triglyceride (TG), nonesterified fatty acid (NEFA), cholesterol, and β-hydroxybutyrate (βHB) levels in the serum of male Rab30KO and littermate control (Ctrl) mice. Fed, $n = 8$/genotype. Fast, $n = 7$/genotype, except $n = 6$ Rab30KO for cholesterol. **b** Fasting TG, NEFA, cholesterol, and βHB levels in the serum of male Rab30$^{L-/-}$ and control (ff) mice. $n = 8$/genotype, except $n = 7$ Rab30$^{L-/-}$ for cholesterol. **c** Fed and fasting TG, NEFA, cholesterol, and βHB levels in the serum of male

Cpt2$^{L-/-}$, DKO, and control (ff;ff) mice. $n = 8$ for all genotypes across both states, except $n = 7$ DKO for cholesterol. In all panels, data are represented as average ±SEM. Significant differences ($p$-value < 0.05) were determined by two-tailed unpaired $t$-test in a and b and by Tukey's multiple comparison's test following one-way ANOVA in c. *$p$ < 0.05; shared letters indicate same significance level. $p$-values, ANOVA tables, and source data for relevant panels are provided in the Source Data file.

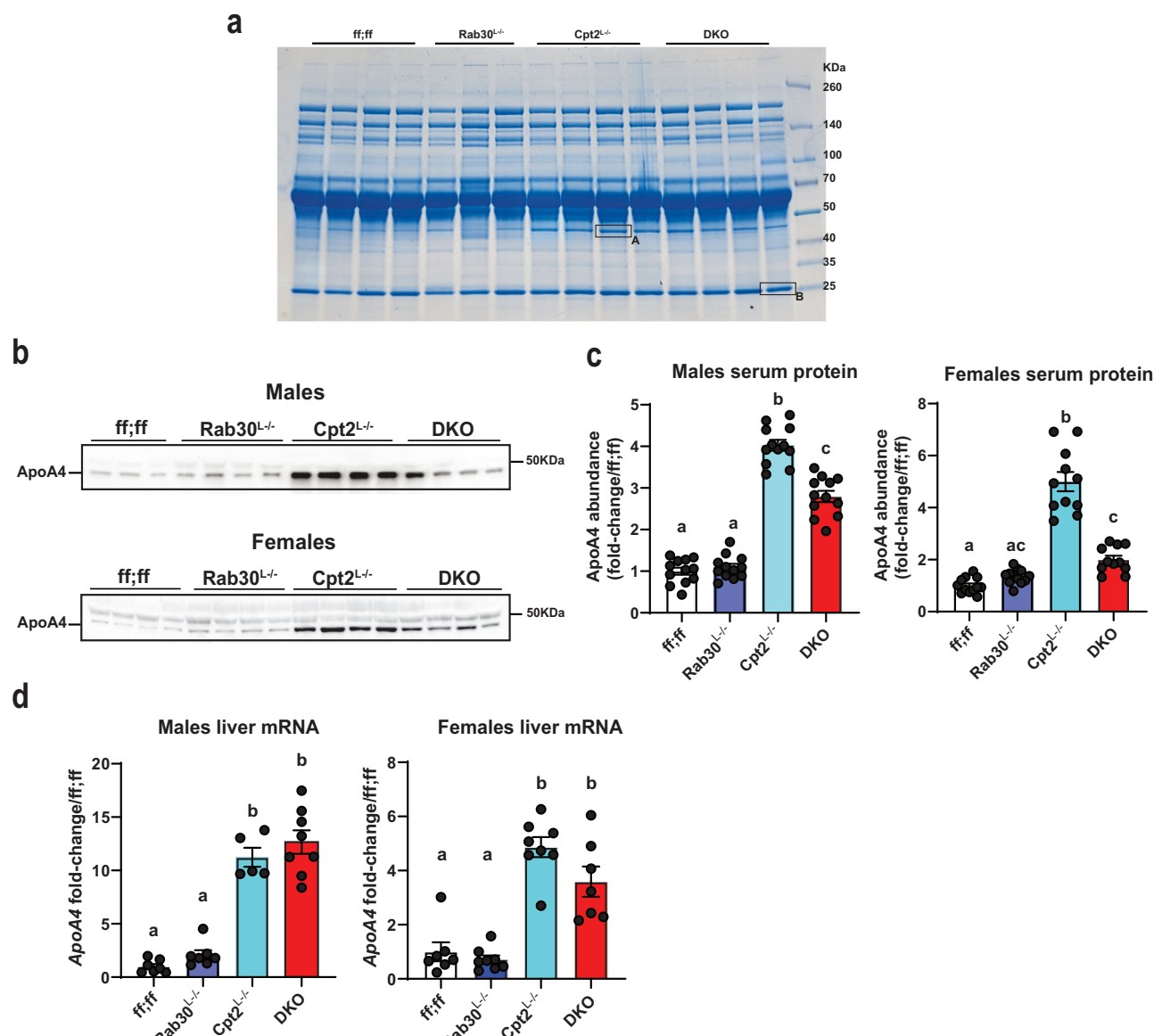

**Fig. 7 | Loss of Rab30 impacts circulating ApoA4 abundance. a** Coomassie stained gel for total protein in the serum (0.5 µl/lane) of fasted male mice. $n = 4$ mice for ff;ff, Cpt2$^{L-/-}$, and DKO, while $n = 3$ mice for Rab30$^{L-/-}$. Bands A and B were excised for mass spectrometry analysis. **b** Representative ApoA4 immunoblot in the serum (1 µl/lane) of 24 h fasted ff;ff, Rab30$^{L-/-}$, Cpt2$^{L-/-}$, and DKO males and females. **c** Left, quantification of ApoA4 band intensity in serum of 24 h fasted males ($n = 12$ males/genotype); Right, quantification of ApoA4 band intensity in serum of 24 h fasted females ($n = 12$ females for ff;ff and Rab30$^{L-/-}$, and $n = 11$ females for Cpt2$^{L-/-}$ and DKO). See Supplementary Fig. 6a–d for associated western blot for the quantitation. **d** Left, qRT-PCR of *ApoA4* mRNA in 24 h fasted livers of males ($n = 7$ for ff;ff and Rab30$^{L-/-}$, 5 for Cpt2$^{L-/-}$, and 8 for DKO); Right, qRT-PCR of *ApoA4* mRNA in 24 h fasted livers of females ($n = 7$ for ff;ff and DKO and 8 for Rab30$^{L-/-}$ and Cpt2$^{L-/-}$). Letters indicate significance groups by Tukey's multiple comparisons test following one-way ANOVA. ANOVA tables and source data for relevant panels are provided as a Source Data file.

homeostasis and efficient lipoprotein transport. There is precedence for Golgi-localized small GTPases and resident proteins to regulate the maturation and secretion of triglyceride-rich lipoprotein particles. For example, a combination of mouse models and cell biological studies have led to the finding that the trans-Golgi localized small GTPase Arfrp1 is responsible for recruiting Arl1, Golgin-245, and Rab2 to the Golgi to mediate lipoprotein lipidation, packaging, and export from both enterocytes and hepatocytes[36,49–51]. It can be hypothesized that Rabs other than Rab2 might be involved in lipoprotein lipidation at the Golgi, as several Rab proteins have been identified to bind to Golgin GRIP proteins directly[31,44]. Of particular interest is the direct association between Golga4/Golgin-245 and Rab30 that we and others[31,44] have observed. The Golgi resident protein Gorasp2, also known as Grasp55, was also identified to be a Rab30 interactor by proximity labeling. Gorasp2/Grasp55 knockout mice display impaired fat absorption and chylomicron secretion following oral olive oil gavage, indicating a function in lipoprotein metabolism in the intestine[37]; while the function of Gorasp2/Grasp55 in lipoprotein secretion from the liver in the fasting state has not been investigated to our knowledge, it may have a conserved role in both tissues. Intriguingly, we identified interactions between Rab30 and Aup1, which appears to regulate apoB stability[52], by proximity labeling, as well as APOB and APOA2 by yeast two-hybrid. Further biochemical and colocalization approaches are required to validate the interaction, but these findings are consistent with a role for Rab30 in lipoprotein trafficking.

We speculate that Rab30 could be a downstream effector of the Arfrp1-Arl1-Golgin cascade and work in parallel with Rab2 in the fasted mouse liver to promote efficient lipoprotein turnover. RNA-seq analysis

of Rab family members in wildtype livers reveals that the Rab2 isoform Rab2a is amongst the most highly expressed Rabs (Fig. 4d). While we have confirmed that Rab30 is induced in the wildtype fasted mouse liver compared to the fed liver by qRT-PCR, its total RNA as measured by RNA-sequencing and protein abundance as determine by mass spectrometry in the liver is low relative to other Rabs, including Rab2a (Fig. 4d, Supplementary Fig. 4a). Perhaps the Pparα-mediated induction of Rab30 expression under fasting conditions is to support the increased demand of fatty acid re-esterification and lipoprotein flux through the liver that is predominantly dependent on the Arfrp1-Arl1-Golgin-245-Rab2 axis. However, it appears that, at least by the sensitivity of our steady state measurements in Rab30$^{L-/-}$ mice, Rab30 is dispensable for maintaining both liver and circulating lipid homeostasis under fasting conditions, possibly due to how far down in the pathway it acts and causing a more subtle phenotype than an earlier regulator, such as Arfrp1. While RNA-seq and proteomics reveals little changes in Rab expression between Rab30 knockouts and controls (Fig. 4c,d, Supplementary Fig. 4a), we do not know if there is a hyper-activation of other Rabs such as Rab2 by shared regulators with Rab30 in a mechanism independent of changes in their expression in Rab30$^{L-/-}$ livers that act to compensate for the loss of Rab30; alternatively, perhaps there is an increased oxidative capacity in the livers of Rab30$^{L-/-}$ during fasting. However, due to (1) the influx of fatty acids and (2) inability to oxidize fats and perform ketogenesis in mice lacking Cpt2$^{L-/-}$, the demand for fatty acid re-esterification and triglyceride export becomes exacerbated, resulting in Rab30 expression to be induced to similar levels as Rab2a (Fig. 4d) and increased reliance on Rab30 function. Therefore, Rab30;Cpt2 DKO mice illuminate a role for Rab30 in contributing to fasting state hepatic lipid transport and homeostasis.

In summary, we describe the role of Rab30 in Pparα-directed lipid and protein homeostasis in the fasted mouse liver. Rab30 marks dynamic membranes at the Golgi and post-Golgi vesicles in the mouse liver. Loss of Rab30 affects the flux of secreted hepatic proteins and prevents hypertriglyceridemia and hypercholesterolemia in the pathologically fatty livers of fasted mice that cannot oxidize fat. Correspondingly, Rab30 is required for efficient trafficking of ApoA4-enriched lipoproteins particles from the fasted liver. Our results indicate that Rab30 supports hepatic lipoprotein secretion and either recycling or export of hepatic secreted proteins in the fasted mouse liver. Furthermore, these data could imply a physiological role for Rab30 in the recycling and redistribution of lipid in the fasting state from adipose tissue through the liver to other highly fatty acid oxidative tissues, such as the heart and skeletal muscle, to use as substrates for energy during nutrient deprivation.

## Methods

### Animal studies
Rab30 floxed mice were generated by The Johns Hopkins University School of Medicine Transgenic Core Laboratory using CRISPR-Cas9 technology. Exon 4 of the Rab30 gene was targeted with 2 guide RNAs: 5'-TCTAGCAAGGCCGTGCTCTTTGG-3' at the intronic region preceding exon 4 and 5'-CCTTCCAAGAATATAGTATAGTT-3' in the intronic region following exon 4. A repair vector was constructed with loxP sites in the intronic regions flanking exon 4. Rab30 whole-body knockout animals were obtained when the exon was deleted. Floxed lines were crossed to a Cre transgenic line under the albumin promoter (JAX stock #003574, ref. 53) to delete Rab30 specifically in hepatocytes. To create Rab30$^{L-/-}$;Cpt2$^{L-/-}$ double knockouts, Rab30 floxed lines were mated to Cpt2$^{L-/-}$ animals[18,54]. The generation of Atgl$^{L-/-}$, Pparα$^{-/-}$, and Cpt2$^{L-/-}$;Pparα$^{-/-}$ have been described elsewhere[20,55]. To generate Fgf21$^{L-/-}$;Cpt2$^{L-/-}$ mice, we bred Fgf21 floxed animals (JAX stock #022361, ref. 56) to Cpt2$^{L-/-}$ mice.

All mice were initially bred and housed long-term in a facility with ventilated racks on a 14 h light/10 h dark cycle at approximately 23 °C and 50% humidity and ad libitum access to a standard rodent chow

(18% protein, 2018SX, Envigo Teklad Diets). For fasting and feeding studies, 8-10week old mice were collected at 3 pm in the 24 h fasted or fed state. For fed state collections, food was removed 4 h prior to dissection to normalize glycemia. Body weight was taken at the time of dissection. Blood glucose measurements were determined by glucometer (NovaMax, Billerica, MA). Isolated serum and tissues were flash-frozen at time of harvest. Experimental details of diet studies in Fig. 1a were described previously[19].

Serum triglycerides were measured by Infinity Triglycerides Stable Liquid Reagent (Thermo Fisher Scientific, TR-22421) with glycerol standard solution (Millepore Sigma, G7793). Serum cholesterol was measured colorimetrically by Total Cholesterol E Kit (FUJIFILM Wako, 999-02601). Nonesterified fatty acids (NEFA) were measured using the HR Series NEFA HR(2) Assay Kit (FUJIFILM WAKO). For ketones, β-hydroxybutyrate was measured using the LiquiColor assay kit (Stanbio Laboratory, 2440-58). Liver triglycerides were extracted with chloroform-methanol via the Folch method as previously described[57,58]. After drying down the chloroform phase, lipids were resuspended in 3:1:1 by volume tert-butanol:methanol:Triton X-100 and assayed with the respective kit and data was acquired using Gen5 1.09 and 3.13 softwares.

For in vivo overexpression studies, adeno-associated viruses with serotype AAV8 and driven under the thyroxine globulin binding promoter at $2 \times 10^{11}$ genome copies were retro-orbitally injected with into mice 1 week prior to experimentation at 8 weeks of age, with exception to Atgl$^{L-/-}$ mice (14 weeks).

All procedures were performed in accordance with the NIH's Guide for the Care and Use of Laboratory Animals and under the approval of the Johns Hopkins School of Medicine Animal Care and Use Committee.

### RNA extraction, quantitative real-time PCR, and RNA-sequencing
Total RNA was extracted from flash frozen tissue using TRIzol reagent (Invitrogen for Thermo Fisher Scientific, 15596026) with additional purification using RNeasy Mini Kit (Qiagen, 74106), as per manufacturers' instructions. Total concentration was measured and copy DNA was generated from 1 µg total RNA using the High-Capacity cDNA Reverse Transcription Kit (Applied Biosystems for Thermo Fisher Scientific, 4368814). Quantitative real-time PCR analyses were carried out on 10 ng of cDNA using SsoAdvanced Universal SYBR Green Supermix (Bio-Rad, 1725271) per manufacturer recommendations with BioRad CFX Manager 2.1 software. RNA was first normalized to CycloA, 18s, or the average of both to generate a ΔCt value and then ΔΔCt was obtained by normalizing data to mean ΔCt of the reference group[59]. For Fig. 1a, the average of 18s and β-actin was used as reference. Primers listed in Supplementary Table 1.

Total liver RNA was submitted to Novogene Corporation Inc. (China & Davis, CA, USA) for library construction, sequencing, and statistical analysis of pairwise comparisons. Four biological replicates were used for each genotype. For comparisons across all genotypes, raw read counts were normalized in R (version 4.1.3) using DESeq2 (3.17)[60]. KEGG and Gene Ontology pathway analysis was performed using the DAVID bioinformatics database. Up- and down-regulated genes and proteins for a given pairwise comparison that were at least changed by 1.2-fold and $p < 0.05$ were submitted for pathway analysis to the DAVID functional annotation tool. Pathway terms are ranked against the Benjamini-adjusted $p$-value generated by the DAVID functional annotation tool.

### SDS-PAGE and immunoblotting
Frozen tissue was homogenized in lysis buffer (50 mM Tris−HCl at pH 7.4, 150 mM NaCl, 1 mM EDTA, 1% Triton X-100, and 0.25% deoxycholate) with Roche PhosSTOP phosphatase inhibitor (Millepore Sigma, 4906837001) and protease inhibitor cocktail (Millipore Sigma, 1836153001) on ice. Proteins were clarified for 20 mins at 12,000 rpm

and 4 °C and supernatants were transferred to new tubes. Total protein content was estimated by Pierce BCA assay (Thermo Fisher Scientific, 23227). Lysates (30-50 μg/20 μL) were boiled in sample buffer (Bio-Rad, 1610747) and β-mercaptoethanol for 8–10 min at 95 °C. For serum samples, 0.5-1 μL serum was loaded per well following boiling in sample buffer. Proteins were separated on Tris-glycine SDS-PAGE gels. For immunoblotting, proteins were transferred onto PVDF membranes, treated with Ponceau S Staining Solution (Cell Signaling Technology, 59803S), and blocked with 5% non-fat milk in TBST for 1 h at room temperature. Membranes were washed in 1xTBST and were incubated overnight in primary antibody diluted in 3% BSA or 5% non-fat milk in TBST at 4 °C. The next day, membranes were washed in 1xTBST and were incubated in secondary antibodies diluted in 5% milk for 1-2 h at room temperature. For antibodies and dilutions, see Supplementary Table 2. Proteins were visualized using the Amersham ECL Select Western Blotting Detection Reagent (Cytiva, RPN2235) or epifluorescence on an Alpha Innotech MultiImage III instrument. Band quantitation was performed using AlphaView Software 1.3.0.6. Protein bands from tissue samples were normalized to intensities of bands from equal protein loading control, Hsc70. For the quantification depicted in Fig. 7 and Supplementary Fig. 6, the molecular weight area surrounding the protein of interest was cut from 3 independent 4–15% SDS-PAGE gels with $n = 4$ serum samples/genotype and transferred on the same membrane prior to immunoblotting, therefore enabling $n = 12$ samples/genotype to be on the same membrane and directly compared for band intensities.

For total protein staining of SDS-PAGE separated serum proteins, gels were incubated with SimplyBlue Safe Stain (Invitrogen for Thermo Fisher Scientific, LC6060) as per manufacturer's instructions. Protein bands of interest were excised from the gel and were placed in methanol-rinsed Eppendorf tubes. Gel pieces were washed twice in 50% ethanol for 10 min each. Gel pieces were stored at −20 °C prior to digestion and peptide identification by mass spectrometry through the Johns Hopkins School of Medicine Mass Spectrometry and Proteomics Facility using their standard procedures.

## Total liver proteome analysis

Frozen tissue was homogenized in lysis buffer (50 mM Tris−HCl at pH 7.4, 150 mM NaCl, 1 mM EDTA, 1% Triton X-100, and 0.25% deoxycholate) with Roche PhosSTOP phosphatase inhibitor (Millepore Sigma, 4906837001) and protease inhibitor cocktail (Millipore Sigma, 11836153001) on ice. Proteins were clarified for 20 min at 12,000 rpm and 4 °C and supernatants were transferred to new tubes. Total protein content was estimated by Pierce BCA assay (Thermo Fisher Scientific, 23227). Samples were submitted to the Johns Hopkins School of Medicine Mass Spectrometry and Proteomics Facility and were reduced with DTT, alkylated with IAA, TCA/Acetone precipitated then digested overnight with Trypsin/LysC at 1:20 enzyme/protein by mass in 100 mM TEAB. Each sample was individually labeled according to manufacturer protocol then reaction was quenched with 8 ul of 5% hydroxylamine. The combined sample was then dried and reconstituted in 10 mM TEAB and injected onto an Agilent LC with fraction collector and separated into 96 fractions over a 105 min gradient. Samples were then concatenated into 24 fractions and each run on a Lumos Orbitrap (Thermo Scientific) mass spectrometer with a 90-min gradient. Total proteome analysis was performed with Proteome Discoverer 2.4.0.305 (Thermo Scientific) and Mascot 2.8 using RefSeq version 204. $p$-values were calculated by ANOVA and Tukey's honest significant difference test post-hoc for individual proteins with biological replicates.

## Primary hepatocyte isolation

Mice were anaesthetized and placed on a hot plate on low setting. The abdomen was decontaminated with ethanol and the abdominal cavity was opened. The suprahepatic inferior vena cava was clamped and the inferior vena cava was cannulated. The portal vein was severed following initiation of perfusion with 0.5 mM EGTA dissolved in 1X PBS. Primary mouse hepatocytes were subsequently isolated based on a previously published protocol[61] with some adjustments. First 0.5 mM EGTA and then 1X PBS were perfused for 5 min at 5 mL/min. In the final perfusion step with liver digest media (Gibco, 17703-34) supplemented with 0.15 mg/mL collagenase (Millepore Sigma C5138), the media was perfused through the liver at a rate of 5 mL/min for the first minute and then the flow rate was turned down to 2 mL/min for the remaining volume. Following perfusion and dissociation of the liver cells on ice in chilled wash media (Gibco, 17704-024) supplemented with 1% penicillin-streptomycin (Gibco, 15140122), hepatocytes were filtered through 70 μm cell strainer into a 50 mL conical tube and pelleted at $50 \times g$ for 5 min at 4 °C. Media was aspirated and the pellet was resuspended in 15 mL cold was media; hepatocytes were again pelleted at $50 \times g$ for 5 min at 4 °C. This wash and spin step was repeated. After aspiration, hepatocytes were resuspended in 25 mL Medium M199 (Gibco, 11150-059) supplemented with 1% penicillin-streptomycin (Gibco, 15140122) and mixed with 2 mL 10X HBSS (Gibco, 14065-056)/18 mL Percoll (GE Healthcare, 17-0891-01), and centrifuged at $300 \times g$ for 10 min at 4 °C to separate live and dead cells. Media with dead cells was aspirated. The hepatocyte pellet was resuspended in Medium M199 supplemented with 10% FBS and 1% penicillin-streptomycin and live cells were quantified via Trypan blue exclusion by hemocytometer. Hepatocytes were plated on home-made collagen-coated 6-well plates at roughly 500,000 cells/well, ethanol and flame-sterilized coverslips in 6-well plates at roughly 500,000 cells/well, or collagen-coated Ibidi 8-well μ-slides at roughly 90,000 cells/well. Cells were allowed to adhere at least 4 h prior to changing media and adding treatments.

For the experiment depicted in Fig. 1e, cells were treated with 10 μM WY-14643 or DMSO in Medium 199 supplemented with 10% FBS and 1% penicillin-streptomycin 100× solution for 16 h. Total RNA was extracted using TRIzol reagent. cDNA was generated and qPCR was performed as mentioned in a previous section.

For experiments depicted in Fig. 2, live primary hepatocytes were stained with Cell Navigator NBD Ceramide Golgi Staining Kit *Green Fluorescence* (AAT Bioquest, 22750) as per manufacturer's instructions. Cells were subjected to live-cell imaging or washed in 1xPBS and fixed in 4% paraformaldehyde for 20 min before imaging.

## Cloning, adenoviral vectors, and stable cell line generation

3XHA-TurboID-Rab30 was made by amplifying Rab30 from EGFP-Rab30 plasmid (Addgene plasmid # 49607) and inserting into 3XHA-TurboID-NLS (Addgene plasmid # 107171) backbone. For mScarletI-Rab30, mScarletI was cloned into the EGFP-Rab30 plasmid by replacing EGFP. TurboID-Cpt1a was generated by PCR amplifying the TurboID (without the 3XHA tag) and ligating into a Cpt1a-containing plasmid using standard cloning procedures. Adeno-associated vectors were constructed by Vector Biolabs (PA, USA) using the AAV8 serotype and TBG promoter for expression.

Stably expressing HA-mScarletI-Rab30 AML12 cells (CRL-2254) were generated using standard cloning procedures. HA-mScarletI-Rab30 PCR product was inserted into pEF6/V5-His TOPO vector by TOPO TA cloning (Invitrogen, K961020). HA-mScarletI-Rab30 plasmid was delivered into AML12 cells via FuGENE HD transfection reagent (Promega, E2311), and vector-expressing colonies were selected by blasticidin resistance 48 h post transfection. AML12 cells were maintained in DMEM/F-12, HEPES (Gibco,11330032) supplemented with 10% BCS, 1% penicillin-streptomycin, 1% ITS (Corning, 25-800-CR), and 35.88 ng/mL dexamethasone at 37 °C; HA-mScarletI-Rab30 stably expressing cells were maintained with an additional 10 μg/mL blasticidin in the media.

## Histology and immunohistochemistry

For liver histology, tissue was fixed in 10% neutral buffered formalin, embedded in paraffin, sectioned, and stained with H&E by The Johns Hopkins Reference Histology Core. For transmission electron microscopy, livers and hippocampi were chopped using the McIlwain Tissue Chopper and fixed in a 2% paraformaldehyde, 2% gluteraldehyde-PBS solution. A 3mm³ block of tissue was further processed for TEM with osmium tetroxide by the Johns Hopkins University School of Medicine Microscope Facility. Ultrathin sections were then cut and imaged with a Hitachi 7600 microscope as previously described[58].

For immunohistochemistry, tissue was fixed in 4% PFA at 4 °C overnight. Tissue was washed with 1xPBS 3 times for 5 mins, followed by incubation in 10% sucrose/1xPBS during the day (~8 h) and then transferred to 30% sucrose/1xPBS overnight. Tissue was frozen in OCT compound and 7 µm sections were cut using a cryostat and dried at room temperature. Slides were processed or frozen at −80. Slides were rehydrated in 1xPBS in 3 by 5 min washes. Tissue sections were surrounded by hydrophobic barrier and blocked in 5% donkey serum/0.5%Triton X-100/1xPBS for 1 h at room temperature in a humidified chamber. Primary antibody was diluted in in blocking buffer and incubated overnight at 4 °C in a humidified chamber. Slides were washed out of primary antibody, and secondary antibody was diluted in blocking buffer for 2 h at room temperature in a humidified chamber. Slides were washed 3 by 5 min in PBST. Slides were incubated in 1:5000 Hoescht 33258 (Invitrogen, H3569) for 5 min. Slides were rinsed in PBST and coverslips were mounted in fluorescent mounting media and sealed with nail polish. For BODIPY 493/503 (Invitrogen, D3922) staining, rehydrated slides were incubated in 0.1 mg/mL BODIPY 493/503 for 30 min in the dark before staining with Hoescht 33258. For antibodies and dilutions, see Supplementary Table 2.

## Confocal microscopy and live-cell imaging

Fixed cell or tissue fluorescent images were acquired using a Zeiss AxioObserver with 880-Quasar laser scanning confocal microscope with a 40x/1.30NA or 63x/1.4NA oil immersion objective and Zen (Zeiss) 2.3 SP1 software. Excitation wavelengths used were 405 nm, 488 nm, and 561 nm. Images were processed in Fiji. Super-resolution live-cell imaging was performed using a Nikon/Yokogawa CSU-W1 SoRa spinning disk confocal microscope with a 60x/1.49NA oil immersion objective. Excitation wavelengths used were 405 nm, 488 nm, and 561 nm. Live-cell imaging data were processed in NIS Elements Viewer 4.50 (Nikon) and Fiji (2.3.051).

## Yeast two-hybrid

The ULTImate Y2H screen was performed by Hybridgenics. *Homo sapiens* RAB30(aa 1–198) was expressed in a pB29 vector encoding a C-terminal LexA DNA binding domain (RAB30(aa 1–198)-LexA fusion) and screened against the Human Liver_RP1 prey library. A total of 77 prey fragments were PCR amplified and sequenced for identification. Gene Ontology pathway analysis for cellular component terms associated with identified interactors was performed using the DAVID bioinformatics database. The data are presented in Supplementary Data 1.

## in vivo TurboID and biotinylated protein enrichment

3XHA-TurboID-Rab30, TurboID-Cpt1a, or EGFP expressing mice were injected either in the fed state or after a 21 h fast with 24 mg/kg bodyweight biotin[62,63]; a 3 h labeling period was allowed prior to dissection. Protein was extracted from flash frozen liver in lysis buffer with protease and phosphatase inhibitors on ice. Total protein of clarified lysates was quantified by BCA, and 5 mg protein was diluted to a final volume of 600 µL in dilution buffer (10 mM Tris-HCl/150 mM NaCl/0.5 mM EDTA with protease and phosphatase inhibitors added). Biotinylated peptides were immunoprecipitated on streptavidin magnetic beads (NEB, S1420S) overnight at 4 °C, rotating end-over-end.

After removing flowthrough, beads were washed 3x in dilution buffer, 3x in 1%SDS, and 3x in 1XPBS, changing tubes every second wash. For western blot analysis of enriched biotinylated proteins depicted in Supplementary Fig. 2, bound proteins were eluted by boiling all or a fraction of the beads in sample buffer (Bio-Rad, 1610747) and β-mercaptoethanol for 10 min at 95 °C followed by magnetic separation of the beads and isolation of the eluted fraction. For protein identification of streptavidin pulldown samples by mass spectrometry, beads were frozen in 1X PBS prior to elution.

## Mass spectrometry analysis of streptavidin pulldown samples following on-bead digestion

100 ul of 100 mM Triethyl Ammonium Bicarbonate (TEAB, pH 8) was added to the solution containing the streptavidin beads, followed by reduction with 50 mM Dithiothreitol at 60 °C for 45 min and subsequent alkylation with 100 mM Iodoacetamide at room temperature in the dark for 15 min. The samples were then digested on the beads by adding 8 ug of Trypsin/LysC (1:50 enzyme:protein) mixed protease (Pierce) and incubating overnight at 37 °C and 1000 rpm. Beads were separated from the digested peptides magnetically. The peptides were further processed to remove detergents and background components using Sera-Mag beads (GE Healthcare GE24152105050350) and standard SP3 peptide cleanup protocols. The TurboID-Rab30 and TurboID-Cpt1a samples were injected into a Thermo Fisher Scientific Orbitrap Fusion Lumos or Q Exactive Plus Orbitrap mass spectrometers, respectively, and eluted over a 90-min gradient from 2% to 90% acetonitrile containing 0.1% formic acid. Mass spectrometry settings were 120,000 resolution for MS1 precursors and 30,000 for MS2 fragment ions with a 3 s cycle time between precursors. AGC for MS2 was set to 200% (1e5) with a maximum injection time of 54 ms and a normalized collision energy of 34.

Raw files obtained from mass spectrometry runs were searched by a label-free quantitation workflow in Proteome Discoverer (Thermo Scientific, version 3.1.0.638) against the Uniprot UP589 *Mus musculus* database using the CHIMERYS identification node prediction model inferys_3.0.0_fragmentation. Oxidation on M and Carbamidomethyl (C) were set by CHIMERYS as dynamic and static modifications, respectively. Supplementary Data 3 provides the total list of proteins identified across all replicates of TurboID-Rab30 and TurboID-Cpt1a streptavidin pulldowns prior to filtering for proteins with at least 2 peptides, removal of contaminants (i.e., keratin), and normalizing intensities to pyruvate carboxylase levels. We normalized to pyruvate carboxylase, a highly abundant and endogenously biotinylated protein, so that we could better compare the TurboID-Rab30 and TurboID-Cpt1a datasets, as they were run on different mass spectrometers. Additionally, statistically significantly enriched proteins were retained only if the proteins were found in at least 2 of the 4 replicates. Supplementary Data 2 displays the proteins identified in replicates of TurboID-Rab30 or TurboID-Cpt1a livers after normalization to pyruvate carboxylase and filtering as described and was used to generate the volcano plot in Fig. 2f. For both Supplementary Data 2 and 3, statistical significance is reported as the adjusted *p*-value using the Benjamini−Hockberg correction for the false discovery rate (FDR). Gene Ontology pathway analysis for cellular component and biological process terms associated with significantly enriched interactors was performed using the DAVID bioinformatics database as cited in the manuscript body.

## Statistical analysis

All data analyses were analyzed using GraphPad Prism software and Microsoft Excel, except for total liver RNA-seq and proteomics, which were addressed in previous sections. Significance was determined using unpaired two-tailed *t*-test or one-way ANOVA with Tukey's post-hoc correction for single or multi-variable expression as specified. For Fig. 1a Welch and Brown−Forsythe ANOVA with Tamhane's T2 multiple

comparisons test *post-hoc* was used due to unequal variances between groups. Data with replicate values are expressed as the mean ± SEM. A $p < 0.05$ was considered significant. For pairwise comparisons, asterisks (*) denote significance levels as specified. For ANOVA, letters represent significance groupings, where shared letters indicate that one group is not significantly different from another group.

## Reporting summary

Further information on research design is available in the Nature Portfolio Reporting Summary linked to this article.

## Data availability

RNA-seq data were deposited in GSE240396. The mass spectrometry proteomics data have been deposited to the ProteomeXchange Consortium via the PRIDE partner repository with the dataset identifier PXD044528 for whole liver proteome and PXD050376 for the proteins identified from TurboID following streptavidin pulldown. Source data are provided with this paper.

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

## Acknowledgements

This work was supported in part by a National Institutes of Health grant R01DK120530 to M.J.W. This material is based upon work supported by the National Science Foundation Graduate Research Fellowship Program under Grant No. DGE1746891 to D.M.S. Any opinions, findings, and conclusions or recommendations expressed in this material are those of the authors and do not necessarily reflect the views of the National Science Foundation. Confocal microscopy images reported in this publication was supported by the Office of the Director and the National Institute of General Medical Sciences of the National Institutes of Health under award number S10OD023548 to Scot C. Kuo of The Johns Hopkins University School of Medicine Microscope Facility and Departments of Biomedical Engineering and Cell Biology. We thank Chip Hawkins in the Transgenic Core Laboratory and Michael Delannoy for producing the TEM images.

## Author contributions

D.M.S. and M.J.W. conceived the research project, interpreted the data, and wrote the manuscript. D.M.S. performed all animal and cell culture experiments; processed samples for proteomics, RNA-seq, qRT-PCR, immunoblotting, biochemical assays, and histology and immunofluorescent staining of OCT embedded tissues; acquired confocal microscopy images; and executed bioinformatic analysis. B.Y.L. assisted with qRT-PCR assays, immunoblotting, and histology and immunofluorescent staining of OCT embedded tissues.

## Competing interests

The authors declare no competing interests.
