## [Peer Review File · Nature Communications]

Rab30 facilitates lipid homeostasis during fastingREVIEWER COMMENTS

Reviewer #1 (Remarks to the Author):

This paper has explored the potential role of RAB30 in the liver. RAB30 was identified as being highly upregulated in the livers of liver-specific CPT2 deficient mice. Based on that observation, the authors decided to examine a possible role of RAB30 in lipid metabolism. While the suggestion that RAB30 may play a role in lipid metabolism is potentially exciting, the evidence presented is preliminary. Overall, the paper raises a lot of questions but unfortunately doesn't really provide clear answers.

Major comments

1) The paper reports on a collection of interesting observations. However, each of these observations remains relatively descriptive and superficial. Further regulatory and especially functional studies are required to better understand the role of RAB30. In the present form, the paper presents bits and pieces that collectively do not give much insight into the likely function of RAB30 in the liver yet provide leads for further study (for example, the suggestion that RAB30 may be linked to lipid excursion, the connection with APOA4, role in protein secretion etc). The claim in the title that RAB30 regulates lipid homeostasis is highly premature considering the depth of the evidence presented.

2) The paper raises a lot of questions but doesn't really provide clear answers. For example, the possible link between RAB30 and APOA4 is interesting but the data are preliminary. It isn't clear why RAB30 deficiency would impact plasma APOA4 levels (but not liver levels) nor is it clear what the functional implications are. It is suggested that APOA4 could be playing a significant role in lipid turnover in the hepatocyte under the pathophysiological conditions of the *Cpt2L^{-/-}* livers but this notion is very speculative. It is even more speculative to argue that RAB30 might act to reduce lipid burden in the liver by promoting the excursion of APOA4-containing lipoproteins. The authors are encouraged to experimentally investigate these hypotheses in more detail.

3) 173 interacting partners for RAB30 is a lot. Performing further experimental validation and functional elaboration on (some of) these interacting partners would greatly strengthen the manuscript.

Minor comments

1) The argument for why the localization of RAB30 was studied in relation to lipid droplets doesn't really make sense. The hepatic lipid content is elevated in *Cpt2L^{-/-}* and DKO livers, yet RAB30 is only increased in the *Cpt2L^{-/-}* mice.

2) In figure 1a, Rab30 mRNA is induced by fasting in WT mice. In figure 1b, Rab30 mRNA is not induced by fasting in WT mice. All six conditions shown in figure 1b should be normalized to one condition (WT fed) to allow for proper interpretation of the data.

3) Please refrain from showing the actual p values in the figures. It is recommended to use asterisks to indicate the P value.

4) Please verify the calculation for the liver triglyceride content. Levels are reported at around 10 mg/g protein. Given that the liver is about 18% protein (wt/wt), this would translate into a fat content of 0.18%, while it should vary from around 1-2% in lean mice in the fed state to well over 5% in 24h fasted mice.

5) Liver-specific deficiency of RAB30, but not whole-body knockout of RAB30, was found to be associated with reduced liver weight. Do the authors have an explanation for these seemingly discrepant observations?

6) Several parameters are shown for male and female mice. However, for many other measurements, it is not clear if these were derived from males, females, or a combination of males and females. This is sometimes (Figure 4D) but often not clarified in the figure legend (for example, figure 3K). The authors might consider restricting the main figures to one particular sex and showing the results of the other sex in the supplement. The way the figures are currently organized between males, females, and unknown is a bit confusing. What further adds to the confusion is the lack of consistency of the effect of RAB30 deficiency in males and females (for example, plasma FGF21, liver weight, blood glucose).

7) Why aren't the data shown in figure 4D presented as in figure 4A (all groups combined in one figure)?

Reviewer #2 (Remarks to the Author):

The manuscript by Smith et al provides new insight into the role of Rab30 in hepatic lipid and protein homeostasis during the fasting state. Using novel mouse models, dietary interventions, different physiological states (i.e., fasting, fast refeeding), and state-of-the-art biochemical and omics approaches, the authors show that: 1) Rab30 is a specific hepatic fasting-induced gene regulated by Ppar α expression, 2) Rab 30 plays a selective role in lipid vectorial secretory pathway transport during fasting, 3) Rab30 interacts with a subset of proteins in the hepatocyte secretory pathway, 4) Rab30 helps mediate the dyslipidemia associated with increased hepatic lipid secretion during the fasting state, and 5) Rab30

promotes secretion of apoA4, as well as other hepatic secretory proteins, during fasting to mitigate hepatic triglyceride accumulation. These data provide novel insights into Rab30 function in vivo and suggest a key role for Rab30 in redistribution of fatty acids fluxing from adipose tissue to the liver to extrahepatic tissues (skeletal and heart muscle) for energy production during fasting. The studies are performed well with multiple complementary approaches and controls. The results support the authors' conclusions and the discussion of results is balanced and comprehensive.

Specific comments-

- 1- Introduction, last paragraph- please clarify "fatty acid deficient livers." This statement does not make sense.
- 2- Figure 1A and B- Is one-way ANOVA appropriate when the standard deviations of the fast and Cpt2^{-/-} groups are so much greater than the other groups?
- 3- Figure 1D- because of the variation in Rab30 protein expression, more samples are needed for Western blots with band intensity analysis.
- 4- Pg 7- "...fasted Rab30 knockout livers as compared to fasted Ppar α or fed wildtype livers." Should be "Ppar α knockout".
- 5- Figure 4C- the hepatic lipid phenotype is subtle and could be demonstrated better with Oil red O staining.

Reviewer #3 (Remarks to the Author):

The manuscript by Smith and colleagues describes a novel role for hepatic Rab30 in lipoprotein trafficking. The work follows up on data from the Wolfgang lab that found Rab30 is induced by loss of CPT-2. Here they show that Rab30 induction occurs through PPAR α action and that it is involved in lipid homeostasis in liver. They also show that while Rab30 associates with the Golgi, it likely does not play a structural role in this interaction. Using Rab30/CPT-2 LDKO mice, they found that Rab30 is necessary for the hyperlipidemia that accompanies loss of CPT2. The study is nicely presented and easy to follow. The data are generally convincing within the scope of the circumstances studied. I have only a few specific comments:

- 1) The manuscript states that: "At the tissue level, the Cpt2L^{-/-} and DKO livers are visibly paler and larger due to fat accumulation than the control and Rab30L^{-/-} livers (Fig. 4B)."

I wonder whether there is an alternative explanation for the liver size phenotype in figure 4? The problem with the increased triglyceride explanation is that the TG assays (female KOs in 4D) don't indicate enough TG to account for the near doubling of liver size (female DKOs in 4A). Granted I'm doing back of the envelope calculations here, but to account for the ~doubling of liver mass would require more than 50% liver TG by mass. The quantification of liver TG is given per g protein, which makes it difficult to translate to liver TG content by mass. The histology shows lipid accumulation, but it doesn't look like 50% liver TG. All that to say, are the authors sure that the liver mass change is due to steatosis? I'm not sure you've done enough to rule out autophagy, though I don't think that's necessary in the context of this manuscript.

2) Given the steatosis phenotypes, Is it possible that some of the TG effects are mediated by hepatic insulin action? Do the various KOs have altered insulin sensitivity?

3) Since both fasting and HFD promotes FFA reesterification (e.g., FA-TG cycle), it would be worth examining the esterification pathway, at least at the expression level (e.g., GPAT, AGPAT, DGAT).

4) Fatty acid reesterification and transport is said to be greatly induced by fasting in mice, yet the single knockouts seem to have a relatively modest phenotype on their own. Is the reesterification and TG secretion pathway much greater in the CPT2 KOs, such the DKO better illuminates the role of Rab30? I think it would help to touch on the importance (or perhaps the dispensibility?) of Rab30 in normal physiology a little more in the discussion.

Reviewer #4 (Remarks to the Author):

As we were specifically asked for evaluating the proximity proteomics experiment, our comments for the technology point of view are as following:

1. As presented in Supplementary Fig. 2, the authors presented the successful biotinylation of many proteins in the total cell lysate and pull-down sample. However, the authors should add the bait protein Rab30 in their wb to properly validate the successful labeling and pull-down of the bait protein. More importantly, the authors failed to present necessary experimental details about how they actually did the pull-down for the wb in Supplementary Fig. 2B.

2. In the method section, the authors described their procedure for TurboID labeling and biotinylated peptide enrichment by following the published work ref. 60. However, the authors didn't explain how they elute the biotinylated peptide for MS analysis which is the most challenging part of this workflow as streptavidin-biotin interaction is really hard to be broken and therefore the recovery of biotinylated could be very low.

3. As presented in Fig. 2F-I and Supplementary Table 2, the authors failed to present the proximity proteomics data properly. As presented in Supplementary Fig. 1, the authors have to present the typical volcano plot with EGFP as control, biological triplicates, and proper statistic evaluation of their data before ending up to the functional annotation of the proteomic data as presented in Fig. 2F. All the identified total proteins, PSMs, biotinylated peptide information have to be included in the Supplementary Table 2. Otherwise, the quality of the proteomics data can not be evaluated.
4. It is interesting that the authors identify biotinylated peptides for the cytoplasmic domains of plasma membrane receptors such as LRP1 and ASGR1. However, the authors didn't present any detailed information about the MS identification and quantification of these biotinylated peptides including PSMs, FDR and relative quantification as compared with their EGFP control.

Introduction: We thank the reviewers for their time and valuable comments. The consensus of the review was that “The studies are performed well with multiple complementary approaches and controls. The results support the authors’ conclusions and the discussion of results is balanced and comprehensive.” We certainly appreciate these comments. There were some minor concerns and we have worked diligently to address these concerns with new experiments and analysis. We believe the reviewers and readers will agree that we have provided a comprehensive analysis of a completely novel component of hepatic lipid homeostasis. Below we address specific comments.

REVIEWER COMMENTS

Reviewer #1 (Remarks to the Author):

This paper has explored the potential role of RAB30 in the liver. RAB30 was identified as being highly upregulated in the livers of liver-specific CPT2 deficient mice. Based on that observation, the authors decided to examine a possible role of RAB30 in lipid metabolism. While the suggestion that RAB30 may play a role in lipid metabolism is potentially exciting, the evidence presented is preliminary. Overall, the paper raises a lot of questions but unfortunately doesn’t really provide clear answers.

Major comments

1) The paper reports on a collection of interesting observations. However, each of these observations remains relatively descriptive and superficial. Further regulatory and especially functional studies are required to better understand the role of RAB30. In the present form, the paper presents bits and pieces that collectively do not give much insight into the likely function of RAB30 in the liver yet provide leads for further study (for example, the suggestion that RAB30 may be linked to lipid excursion, the connection with APOA4, role in protein secretion etc). The claim in the title that RAB30 regulates lipid homeostasis is highly premature considering the depth of the evidence presented.

Thank you for the comment. We agree that there is a lot more to be investigated in regard to Rab30’s role in lipid metabolism. While some work has been performed by others using cell culture *in vitro* to describe putative roles for Rab30 in Golgi structural maintenance,^{1,2} transport of an integral membrane protein between the Golgi and plasma membrane via endosomes,² integrin trafficking,³⁻⁵ and pathogen invasion,^{6,7} to name a few, our manuscript is the first direct

interrogation of the **physiological** impact of hepatic Rab30 expression. Previously, there have only been the aforementioned cell culture studies, which provided little if any insight into Rab30 function at the organismal level, and vague correlations between Rab30 expression, liver function under metabolic duress, and Ppara activity. Here, we have used multiple orthogonal unbiased methods and rigorous mouse models to investigate liver function impacted by Rab30 and have defined unique physiological roles for Rab30 in managing lipid homeostasis.

Specifically:

- We have established 3 mouse lines specifically for this paper. By creating a DKO using the Cpt2-null background to endogenously amplify the effect of the loss of Rab30 in the mouse liver, we were able to find a condition that allowed us to study Rab30 function *in vivo* and uncover clear disruptions in circulating triglyceride levels, ApoA4 levels, and a disruption in hepatic abundance of secreted proteins and proteins in the endolysosomal pathway in the absence of Rab30. While the KO has a phenotype of diminished lipid excursion, the double KO greatly exaggerated, solidified, and expanded these data. Many experiments using mouse models fail to find the appropriate physiologic conditions that elicit a phenotype. Due to the tenacity of the graduate student working on this project, we were able to define the conditions that elicited a robust *in vivo* phenotype regulated by this enigmatic Rab protein.
- We have used both yeast two-hybrid and *in vivo* proximity labeling to determine Rab30 interactors in hepatocytes, which has enabled us to confirm prior interactions from other cell culture systems and to establish new targets in hepatocytes. Follow-ups to the functional output of these interactions will allow for mechanistic determination of the types of cargo Rab30 is involved in trafficking.
- We have interrogated the role of Rab30 in Golgi structure in mice. Golgi structure appears to be altered by siRNA knockdown of Rab30 in cultured cells,^{1,2} but not in CRISPR/Cas9 edited MDCK cells.⁸ However, the Golgi dispersion phenotype has predominated the field and led to the overall and widely cited conclusion that Rab30 is Golgi resident and required for the structural maintenance of Golgi morphology. In fact, the original paper demonstrating the Golgi dispersion phenotype, published in 2012¹, has been cited over 30 times. However, our more stringent method of genetic manipulation of Rab30 by CRISPR/Cas9 coupled with TEM and live cell super-resolution microscopy shows that the Golgi structure is unaltered by Rab30 knockout *in vivo* (Fig.

3f,g), which leads to an entirely different interpretation of the reported role for Rab30 in the maintenance of Golgi ultrastructure.

- We have characterized Rab30 localization in the fasted mouse liver and how its localization may be altered due to hepatic Ppar α activation using different mouse models with genetic manipulations in β -oxidation (Cpt2^{L-/-}) or in triglyceride hydrolysis (Atgl^{L-/-}).
- We have investigated Rab30 function in several different Ppar α -directed liver processes, such as autophagy, hepatic triglyceride storage, and circulating triglyceride homeostasis.

Finally, we think the title “Rab30 regulates lipid homeostasis in the hepatocyte during fasting” is appropriate but would certainly be open to suggestions.

2) The paper raises a lot of questions but doesn't really provide clear answers. For example, the possible link between RAB30 and APOA4 is interesting but the data are preliminary. It isn't clear why RAB30 deficiency would impact plasma APOA4 levels (but not liver levels) nor is it clear what the functional implications are. It is suggested that APOA4 could be playing a significant role in lipid turnover in the hepatocyte under the pathophysiological conditions of the Cpt2L-/- livers but this notion is very speculative. It is even more speculative to argue that RAB30 might act to reduce lipid burden in the liver by promoting the excursion of APOA4-containing lipoproteins. The authors are encouraged to experimentally investigate these hypotheses in more detail.

Thank you for your comment. We agree with your sentiments and would like to understand more about how the effectors we identified lead to altered cargo export. However, to reiterate the above point, this paper is the first to explore the function of Rab30 *in vivo* and to characterize its role as a Ppar α target gene in the fasted mouse liver. We have several hypotheses of how Rab30 may be functionally linked to triglyceride secretion. However, these hypotheses would require a substantial amount of experimentation including defining the role of ApoA4 which has been itself controversial for many years. We believe these experiments are outside the intended scope of our current data intense manuscript, and are instead best left to be addressed more thoroughly in follow up studies.

A final note to address the concern on the putative role of ApoA4 expression in the liver: Our suggestion that ApoA4 may be playing a significant role in lipid turnover in the pathological conditions of the Cpt2^{L-/-} liver is indeed speculative, as ApoA4 is itself another protein with an unclear function in liver physiology. While ApoA4 expression in the post-prandial intestine is well characterized, the conditions for its regulation in the liver are still being uncovered. There is

evidence for the transcriptional regulation of ApoA4 by *Erra*, *Pgc1 α* , *Hnf4 α* , and even *Ppara*, but it appears that *Crebh* and *Atf3* are the strongest regulators of ApoA4 expression *in vivo*, including in the fasting liver and in under times of metabolic duress and fatty liver, such as following high fat diet feeding.⁹⁻¹² As mentioned in the Discussion, the suggestion that ApoA4 promotes lipid excursion from the liver comes from overexpression studies in which ApoA4 expression correlated with increased VLDL secretion from the liver.¹³ However, much remains to be investigated on the role and regulation of ApoA4 in maintaining lipid homeostasis in the liver, specifically in regards to triglyceride secretion from the fatty livers, and along with furthering a mechanistic understanding of Rab30's role in this process, this again cannot all be addressed in one manuscript.

3) 173 interacting partners for RAB30 is a lot. Performing further experimental validation and functional elaboration on (some of) these interacting partners would greatly strengthen the manuscript.

We agree that 173 *direct* interacting partners for Rab30 is a lot. However, the TurboID approach used defines an interactor network that is not only comprised of direct interactors, but also proteins in proximity to the TurboID-fusion protein, which is why the list of interactors is longer than would be expected for direct interactors of Rab30. As the Golgi is the hub of the secretory pathway and as dynamic vesicles budding from the Golgi reach other compartments of the cell, it actually seems likely that we would find a large number of proteins in the Rab30 interactome. We also agree that functional elaboration of some of Rab30 interacting partners would allow us to further define the cellular mechanism of Rab30 action in the liver, and this has already been done for some of known Rab30 partners in cell culture by others. For example, in HeLa cells, Rab30 has been observed to be involved in retrograde trafficking of the human ortholog of one of our putative interactors, *Tgoln1*, from endosomes to the plasma membrane.² However, as we have tried to stress in this manuscript, there is a specific function of Rab30 in the fasted mouse liver that is highly dependent on the metabolic status of the mouse, and using a cell culture model system to verify and annotate the function of hepatocyte-specific Rab30 interactors may not allow us to uphold the physiological conditions of fasting. This question would best be addressed by creating a knock-in mouse model in which the endogenous locus of the *Rab30* gene has been tagged, enabling us to use the handle to perform pull-downs for direct interactors or to isolate Rab30-marked vesicles from the liver. We are particularly interested in the interactions between Rab30 and the E3 ubiquitin ligase *Amfr*, the fatty acid transporters and plasma membrane receptors identified, and the GRIP domain Golgi localized proteins, as there

is precedence for an ARFRP1-ARL1-Golgin-Rab cascade in regulating lipoprotein transport in the mouse liver (See the response to Reviewer #3, point 4 for more information).

Minor comments

1) The argument for why the localization of RAB30 was studied in relation to lipid droplets doesn't really make sense. The hepatic lipid content is elevated in Cpt2L^{-/-} and DKO livers, yet RAB30 is only increased in the Cpt2L^{-/-} mice.

One issue that we feel we need to address is the effect of defective fatty acid oxidation vs. simply having a fatty liver. i.e. what responses are due to loading a liver with lipid droplets compared to specific effects of a block in fatty acid oxidation. That is why we looked at other models of fatty liver such as AtgIL^{-/-} liver.

2) In figure 1a, Rab30 mRNA is induced by fasting in WT mice. In figure 1b, Rab30 mRNA is not induced by fasting in WT mice. All six conditions shown in figure 1b should be normalized to one condition (WT fed) to allow for proper interpretation of the data.

In the original manuscript, for Fig. 1b, all 6 conditions were normalized to WT fed, as depicted both on the Y-axis title and in the figure legend. However, we felt that some aspects of the original Fig. 1 were redundant from panel to panel, and in the newest version of the manuscript, we have changed Fig. 1 to be more concise and combined the data from what was originally Fig. 1b-c after re-performing qRT-PCR analysis.

3) Please refrain from showing the actual p values in the figures. It is recommended to use asterisks to indicate the P value.

Thank you for your recommendation. We changed the actual *p*-values to asterisks in the figures.

4) Please verify the calculation for the liver triglyceride content. Levels are reported at around 10 mg/g protein. Given that the liver is about 18% protein (wt/wt), this would translate into a fat content of 0.18%, while it should vary from around 1-2% in lean mice in the fed state to well over 5% in 24h fasted mice.

Thank you for your attention to detail; the calculations for liver triglyceride content initially reported were indeed incorrect and have been fixed in the newest version of the manuscript.

5) Liver-specific deficiency of RAB30, but not whole-body knockout of RAB30, was found to be associated with reduced liver weight. Do the authors have an explanation for these seemingly discrepant observations?

We thank the reviewer for their comment. We, however, do not view these as necessarily discrepant observations, but perhaps a question of biological significance. These are very small changes that are statistically significant due to the large number of mice we included in our analysis. The average g liv/kg bodyweight difference between the LKO and ff is 1.93g/kg bodyweight (1.93mg/g) (12.99 for LKO and 14.92). The difference in the KO vs control is 1.39mg/g, with the tendency of the KO to be larger (16.78 for KO vs. 15.39 for control), but this difference is not statistically different. The delta between these differences is ~0.55mg/g. The question now becomes is the 1.93mg/g difference between the LKO and ff mice is inherently biologically relevant vs the 1.39mg/g difference in the KO and control mice.

6) Several parameters are shown for male and female mice. However, for many other measurements, it is not clear if these were derived from males, females, or a combination of males and females. This is sometimes (Figure 4D) but often not clarified in the figure legend (for example, figure 3K). The authors might consider restricting the main figures to one particular sex and showing the results of the other sex in the supplement. The way the figures are currently organized between males, females, and unknown is a bit confusing. What further adds to the confusion is the lack of consistency of the effect of RAB30 deficiency in males and females (for example, plasma FGF21, liver weight, blood glucose).

Thank you for your suggestion. In the final manuscript, we have decided to leave both the male and the female data in the main figure for transparency reasons, but the sexes have been declared in the text and/or figure legend. We think that some of the data could indicate a sexual dimorphism in certain aspects of the fasting response that appear to be potentiated with the loss of Rab30. We have not followed up on some of these observations because they were not relevant to our main hypothesis that the secretion or recycling of lipid particles and proteins might be altered with the loss of Rab30. Therefore, we have removed them from the main manuscript but have included the data instead in Supplementary Figure 3.

7) Why aren't the data shown in figure 4D presented as in figure 4A (all groups combined in one figure)?

The inconsistency in the formatting of Figure 4D in comparison to Figure 4A was an error in the preparation of the manuscript. This has been corrected in the latest revision of the manuscript.

Reviewer #2 (Remarks to the Author):

The manuscript by Smith et al provides new insight into the role of Rab30 in hepatic lipid and protein homeostasis during the fasting state. Using novel mouse models, dietary interventions, different physiological states (i.e., fasting, fast refeeding), and state-of-the-art biochemical and omics approaches, the authors show that: 1) Rab30 is a specific hepatic fasting-induced gene regulated by Ppar α expression, 2) Rab 30 plays a selective role in lipid vectorial secretory pathway transport during fasting, 3) Rab30 interacts with a subset of proteins in the hepatocyte secretory pathway, 4) Rab30 helps mediate the dyslipidemia associated with increased hepatic lipid secretion during the fasting state, and 5) Rab30 promotes secretion of apoA4, as well as other hepatic secretory proteins, during fasting to mitigate hepatic triglyceride accumulation. These data provide novel insights into Rab30 function in vivo and suggest a key role for Rab30 in redistribution of fatty acids fluxing from adipose tissue to the liver to extrahepatic tissues (skeletal and heart muscle) for energy production during fasting. The studies are performed well with multiple complementary approaches and controls. The results support the authors' conclusions and the discussion of results is balanced and comprehensive.

Specific comments-

1- Introduction, last paragraph- please clarify "fatty acid deficient livers." This statement does not make sense.

Thank you for the comment. The sentence has been corrected to read "fatty acid oxidation deficient livers."

2- Figure 1A and B- Is one-way ANOVA appropriate when the standard deviations of the fast and Cpt2 $^{-/-}$ groups are so much greater than the other groups?

Thank you for your comment and for helping us improve the rigor of our statistical analyses. Fig. 1B has been revised in the newest version of the manuscript, and Levene's test for homogeneity of variances revealed no difference in the standard deviations. For Fig. 1A, Levene's test revealed a statistically significant difference in the variances between groups.

Data was log transformed, and ANOVA was repeated, yielding the same results between conditions as originally assessed in the manuscript (see graph below). While we presented the RNA-seq data in this way, we prefer not to represent the expression data derived from qRT-PCR as \log_2 transformed for consistency reasons across graphs. Therefore, we have reanalyzed the data represented in Fig. 1A using the Welch and Brown-Forsythe ANOVA with Tamhane's T2 multiple comparison's test and is now reported in the figure legend and methods section. None of the statistical differences or interpretations of the data have changed.

3- Figure 1D- because of the variation in Rab30 protein expression, more samples are needed for Western blots with band intensity analysis.

The original western blot in Fig. 1D has been replaced with larger number of samples for the genotypes with the most variability originally depicted (i.e., wildtype and $Cpt2^{L/-}$), along with the corresponding band intensity analysis for the wildtype and $Cpt2^{L/-}$ animals.

4- Pg 7- “.....fasted Rab30 knockout livers as compared to fasted Ppara or fed wildtype livers.” Should be “Ppara knockout”.

Thank you for your attentiveness; the sentence has been corrected.

5- Figure 4C- the hepatic lipid phenotype is subtle and could be demonstrated better with Oil red O staining.

Thank you for your suggestion. To better demonstrate the hepatic lipid phenotype, we have included BODIPY stained sections of the livers of the various knockouts to Figure 5c. We have also included the Oil Red O staining in Supplementary Fig. 3.

Reviewer #3 (Remarks to the Author):

The manuscript by Smith and colleagues describes a novel role for hepatic Rab30 in lipoprotein trafficking. The work follows up on data from the Wolfgang lab that found Rab30 is induced by loss of CPT-2. Here they show that Rab30 induction occurs through PPAR α action and that it is involved in lipid homeostasis in liver. They also show that while Rab30 associates with the Golgi, it likely does not play a structural role in this interaction. Using Rab30/CPT-2 LDKO mice, they found that Rab30 is necessary for the hyperlipidemia that accompanies loss of CPT2. The study is nicely presented and easy to follow. The data are generally convincing within the scope of the circumstances studied. I have only a few specific comments:

1) The manuscript states that: "At the tissue level, the Cpt2 $^{-/-}$ and DKO livers are visibly paler and larger due to fat accumulation than the control and Rab30 $^{-/-}$ livers (Fig. 4B)."

I wonder whether there is an alternative explanation for the liver size phenotype in figure 4? The problem with the increased triglyceride explanation is that the TG assays (female KOs in 4D) don't indicate enough TG to account for the near doubling of liver size (female DKOs in 4A). Granted I'm doing back of the envelope calculations here, but to account for the ~doubling of liver mass would require more than 50% liver TG by mass. The quantification of liver TG is given per g protein, which makes it difficult to translate to liver TG content by mass. The histology shows lipid accumulation, but it doesn't look like 50% liver TG. All that to say, are the authors sure that the liver mass change is due to steatosis? I'm not sure you've done enough to rule out autophagy, though I don't think that's necessary in the context of this manuscript.

Thank you for your comment. First, just to clarify, the TG assays depicted in what was originally Fig. 4D were performed in male livers. Additionally, our initial calculations for the TG in the livers were incorrect and have been fixed in the newest version of the manuscript. In Cpt2 $^{-/-}$ knockouts, our fasting-induced steatosis phenotypes are quite dramatic, as presented here and previously from our lab.¹⁴ We do not see any difference between littermates and Cpt2 $^{-/-}$ or DKO in terms of fed state large liver lobe/body mass ratio or in H&E stained sections of livers

(Supplementary Fig. 3). In addition to the H&E liver sections, we have also included in the newest version of the manuscript BODIPY (in Fig. 5c) and Oil-Red O (Supplementary Fig. 3) stained liver sections to more definitively show the hepatic lipid accumulation and loss of the zonation in mice lacking Cpt2 in the fasted state. We believe that these data suggest that most, if not all, of the increase in liver/body mass ratio is due to hepatic lipid accumulation.

It is possible that autophagic flux is more altered in the context of our knockouts; however, in addition to the steady state western blot depicted in Fig. 3, qPCR of various autophagy related genes reveals slight increases in Cpt2 and DKO livers compared to wildtype, but minimal difference with the loss of Rab30 (Supplementary Fig. 4). It is also possible that, in the Rab30 knockouts, the membrane for phagophore formation is sequestered more from the ER or organelle membranes other than the Golgi or that Rab30 acts in later stages of autophagy (such as in autophagosome fusion with the lysosome), which is why we do not observe an alteration in lipidation of LC3 levels with the loss of Rab30. However, these hypotheses would have to be more rigorously tested and we agree that further experimentation on this front is also outside the context of the manuscript given the findings of the physiological relationship between Rab30 and fasting lipid levels.

2) Given the steatosis phenotypes, Is it possible that some of the TG effects are mediated by hepatic insulin action? Do the various KOs have altered insulin sensitivity?

Despite not having performed a glucose or insulin tolerance test for the knockouts described in this manuscript, several lines of evidence lead us to believe that the TG effects are not mediated by altered hepatic insulin sensitivity. Our lab has previously shown that fed and fasting blood glucose and insulin levels are not different between Cpt2^{L-/-} and littermate controls.¹⁴ Additionally, our lab has reported that Cpt2^{L-/-} mice are protected from high fat diet induced obesity and exhibit improved insulin sensitivity and glucose tolerance, despite developing a fatty liver and exhibiting systemic serum dyslipidemia.¹⁵ We believe that this phenotype would be driven by loss of Cpt2 in the DKO animals as well.

In the first version of this manuscript, we have shown that males of all genotypes were able to maintain glycemia after a 24hr fast (Fig. 3G); in the newest version of the manuscript, we have included fed state blood glucose measurements (Supplementary Fig. 3b), which are also not altered by genotype compared to littermate controls.

Finally, to directly examine insulin sensitivity in our models, we analyzed insulin signaling by examining phosphorylation of Akt in the fasted livers of male ff;ff, Rab30^{L-/-}, Cpt2^{L-/-}, and DKO mice (A). Insulin signaling results in the phosphorylation and activation of Akt. Quantitation of

total and phosphorylated Akt levels reveals no statistical difference between genotypes in the ratios of Akt phosphorylated at T308 or S473 over total Akt levels (B), despite a downward trend of total and phosphorylated Akt levels in the *Cpt2*^{L-/-} and DKO livers (C), especially the mTOR-dependent phosphorylation site S473,¹⁶ which could be indicative of a slight decrease in insulin signaling and rapamycin-insensitive mTOR complex activity with loss of hepatic fatty acid oxidation.

3) Since both fasting and HFD promotes FFA reesterification (e.g., FA-TG cycle), it would be worth examining the esterification pathway, at least at the expression level (e.g., GPAT, AGPAT, DGAT).

Thank you for your suggestion. We have performed qRT-PCR in the fasted livers of male mice for genes involved in the re-esterification pathway (see graphs below). Out of the triglyceride lipases examined, *Atgl*, *Mgl*, and *Abhd6* are all upregulated 2-3 fold in the *Cpt2*^{L-/-} and *Rab30*;*Cpt2* DKO livers, but *Hsl* is suppressed in the DKO livers compared to control livers. *Acss2* is slightly suppressed in both the *Cpt2*^{L-/-} and DKO livers, while *Gpat1* and *Dgat1* are both induced compared to control. There is a slight but significant decrease in the *Gpat1* and *Dgat1* mRNA in the DKO livers compared to *Cpt2*^{L-/-}. *Dgat2* mRNA remains unchanged across all genotypes examined. The most abundant *Agpat* family members in the liver were also

analyzed. *Agpat2* is slightly decreased in Rab30^{L-/-}, Cpt2^{L-/-}, and DKO livers compared to control. *Agpat3* mRNA is ~1.5-fold increased in Cpt2^{L-/-} and DKO livers. *Agpat5* was also increased in both the Cpt2^{L-/-} and DKO livers compared to control, with there being a slight decrease in mRNA in the DKO compared to Cpt2^{L-/-}. *Agpat4* mRNA is induced only in Cpt2^{L-/-} livers with a notable failure to be induced in the DKO livers, and it is also slightly suppressed in the Rab30^{L-/-} livers. Overall, it appears that (1) genes involved in the FA-TG cycle are largely increased in the fasted DKO and Cpt2^{L-/-} livers compared to control or Rab30^{L-/-} livers, which correspond to our observations of fasting-induced fatty livers and alterations in serum triglyceride levels in DKO and Cpt2^{L-/-} mice, and (2) that the Rab30^{L-/-} and DKO phenocopy the control and Cpt2^{L-/-} transcriptional activity, respectively. It seems unlikely that slight differences between Cpt2^{L-/-} and DKO mice in the transcriptional regulation of these genes involved in the FA-TG cycle would correspond to large changes in the functional output of the cycle as a whole, especially given that total liver triglycerides are unchanged between the two genotypes. However, it is curious that we observe increased NEFA in the serum of the DKO mice, but we have not yet determined if lipolysis is increased in the adipose tissue of the DKO mice.

4) Fatty acid reesterification and transport is said to be greatly induced by fasting in mice, yet the single knockouts seem to have a relatively modest phenotype on their own. Is the reesterification and TG secretion pathway much greater in the CPT2 KO, such the DKO better illuminates the role of Rab30? I think it would help to touch on the importance (or perhaps the dispensability?) of Rab30 in normal physiology a little more in the discussion.

We thank the reviewer for the thought-provoking comment and agree that further elaboration on the importance and/or dispensability of Rab30 in normal physiology in the Discussion is useful for the readers. We do agree with your assessment that the hepatic reesterification and TG secretion pathway is heightened in the Cpt2^{L-/-} livers during fasting, as the liver is not able to dispose of excess fat or provide ketones as energetic substrates for peripheral tissues. Having this extreme metabolic state in the DKO background has indeed helped to illuminate a role for Rab30 in this TG secretion pathway, despite its dispensability during fasting in livers capable of performing fatty acid oxidation. Perhaps other factors involved

in triglyceride secretion are upregulated with the loss of Rab30 in single liver knockouts that we failed to detect by our methods; or, perhaps a slightly increased rate of hepatic oxidation compared to wildtype animals to compensate for the decreased ability of export, despite the fact that total serum triglyceride levels are not statistically altered between the two genotypes. These possibilities remain to be tested. The following response has also been added to the Discussion section in the newest version of the manuscript—please note that the citations are the same here as presented in the text, but the numbering is different:

Based on our current data, we propose a physiological role for Rab30 during fasting in the repackaging of adipose-derived lipids in the liver and their redistribution as lipoproteins to supply other organs with an energy source during fasting. While the assembly of lipoproteins is initiated at the ER, particles are further modified in the Golgi, making it an important organelle for maintaining cholesterol homeostasis and efficient lipoprotein transport. There is precedence for the action of small GTPases at the trans-Golgi to regulate the maturation and secretion of triglyceride-rich lipoprotein particles. For example, a combination of mouse models and cell biological studies have led to the finding that the trans-Golgi localized small GTPase Arfrp1 is responsible for recruiting Arl1, Golgin-245, and Rab2 to the Golgi to mediate lipoprotein lipidation, packaging, and export from both enterocytes and hepatocytes.¹⁷⁻²⁰ It can be hypothesized that Rabs other than Rab2 might be involved in lipoprotein lipidation at the Golgi, as several Rab proteins have been identified to bind to Golgin GRIP proteins directly.^{21,22} Of particular interest is the direct association between the aforementioned Golgin, Golga4/Golgin-245, and Rab30 that was detected both *in vitro* by yeast two-hybrid and in *Drosophila* cell lysates.^{21,22} Again, while we did not detect an interaction with Golga4/Golgin-245 in our yeast two-hybrid screen of GTP-locked Rab30 against a human liver cDNA library, we did find GOLGA5 by yeast two-hybrid, as well as observe an *in vivo* interaction between Rab30 and both Golga4 and Golga5 by proximity labelling; additionally, while not significantly enriched compared to TurboID-Cpt1a samples, Arl1 was also identified in at least two replicates of TurboID-Rab30 samples (**Supplementary Tables 2-4**).

Based on our findings and those previously mentioned regarding the small GTPase cascade regulating lipoprotein lipidation in the mouse intestine and liver, we speculate that Rab30 could be a downstream effector of the Arfrp1-Arl1-Golgin cascade and work in parallel with Rab2 in the fasted mouse liver to promote efficient lipoprotein turnover. RNA-seq analysis of Rab family members in wildtype livers reveals that the Rab2 isoform Rab2a is amongst the most highly expressed Rabs (**Fig. 4d**). While we have confirmed that Rab30 is induced in the wildtype fasted mouse liver compared to the fed liver by qRT-PCR, its total RNA as measured by RNA-sequencing and protein abundance as determined by mass spectrometry in the liver is low relative to other

Rabs, including Rab2a (**Fig. 4d, Supplementary Fig. 4a**). Perhaps the Ppara-mediated induction of Rab30 expression under fasting conditions is to support the increased demand of fatty acid re-esterification and lipoprotein flux through the liver that is predominantly dependent on the Arfrp1-Arl1-Golgin-245-Rab2 axis. However, it appears that, at least by the sensitivity of our steady state measurements in Rab30^{L-/-} mice, Rab30 is dispensable for maintaining both liver and circulating lipid homeostasis under fasting conditions, possibly due to how far down in the pathway it acts and causing a more subtle phenotype than an earlier regulator, such as Arfrp1. While RNA-seq and proteomics reveals little changes in Rab expression between Rab30 knockouts and controls (**Fig. 4c,d, Supplementary Fig. 4a**), we do not know if there is a hyper-activation of other Rabs such as Rab2 by shared regulators with Rab30 in a mechanism independent of changes in their expression in Rab30^{L-/-} livers that act to compensate for the loss of Rab30; alternatively, perhaps there is an increased oxidative capacity in the livers of Rab30^{L-/-} during fasting. However, due to (1) the influx of fatty acids and (2) inability to oxidize fats and perform ketogenesis in mice lacking Cpt2^{L-/-}, the demand for fatty acid re-esterification and triglyceride export becomes exacerbated, resulting in Rab30 expression to be induced to similar levels as Rab2a (**Fig. 4d**) and increased reliance on Rab30 function. Therefore, Rab30;Cpt2 DKO mice illuminate a role for Rab30 in contributing to fasting state hepatic lipid transport and homeostasis.

Reviewer #4 (Remarks to the Author):

Broadly, our TurboID workflow was to identify proteins differentially marked by Rab30 and then to site map a subset of these proteins as validation. We have found that the site mapping of the biotinylated peptides to be an excellent validation of the proteins identified in the pulldown.

As we were specifically asked for evaluating the proximity proteomics experiment, our comments for the technology point of view are as following:

1. As presented in Supplementary Fig. 2, the authors presented the successful biotinylation of many proteins in the total cell lysate and pull-down sample. However, the authors should add the bait protein Rab30 in their wb to properly validate the successful labeling and pull-down of the bait protein. More importantly, the authors failed to present necessary experimental details about how they actually did the pull-down for the wb in Supplementary Fig. 2B.

The western blot for the self-biotinylation of bait protein TurboID-Rab30 has been added to Supplementary Fig. 2. Successful labeling and pulldown of the bait protein(s) is also demonstrated by mass spectrometry identification following on-bead digestion as depicted in Supplementary Fig. 2d (See point 3). The experimental details for the pulldown protocol have

been added to the methods section "*In vivo* TurboID and biotin-site mapping.": "For western blot analysis of enriched biotinylated proteins depicted in **Supplementary Fig. 2**, bound proteins were eluted by boiling all or a fraction of the beads in sample buffer (Bio-Rad, 1610747) and β -mercaptoethanol for 10mins at 95°C followed by magnetic separation of the beads and isolation of the eluted fraction."

2. In the method section, the authors described their procedure for TurboID labeling and biotinylated peptide enrichment by following the published work ref. 60. However, the authors didn't explain how they elute the biotinylated peptide for MS analysis which is the most challenging part of this workflow as streptavidin-biotin interaction is really hard to be broken and therefore the recovery of biotinylated could be very low.

Thank you for pointing out the absence of the experimental protocol in our methods section. We have updated the section to include this information, which is also posted below: "Biotinylated peptides bound to streptavidin linked magnetic beads were separated from the final wash buffer by placing the Eppendorf tubes on a magnetic holder. After removing and setting aside the final wash buffer, the following steps were performed in the fume hood. Neat HFIP (400ul Millipore/Sigma Cat#105228) was added to each tube put on a shaker at 1000 rpm for 10 minutes, then placed on the magnetic holder to recover the HFIP eluted peptides from the magnetic beads. HFIP extraction was performed twice and both peptide elutions were combined in a new Eppendorf tube. The combined extractions were evaporated to dryness in a speedvac and resuspended in 0.1% TFA. Samples were buffer exchanged using an Oasis HLB (Waters, Milford MA) with two washes of 0.1% TFA and eluted with 0.1% TFA, 60% acetonitrile into new tubes and dried by vacuum centrifugation prior to mass spectrometry analysis."

3. As presented in Fig. 2F-I and Supplementary Table 2, the authors failed to present the proximity proteomics data properly. As presented in Supplementary Fig. 1, the authors have to present the typical volcano plot with EGFP as control, biological triplicates, and proper statistic evaluation of their data before ending up to the functional annotation of the proteomic data as presented in Fig. 2F. All the identified total proteins, PSMs, biotinylated peptide information have to be included in the Supplementary Table 2. Otherwise, the quality of the proteomics data cannot be evaluated.

We sincerely apologize for the missing information in the original version of the manuscript. The information for total proteins found from the biotin site-mapping experiment (prior to filtering for the biotin moiety) is presented in **Supplementary Table 8**. The requested

information for PSMs and biotinylated peptide information has been included in an independent table, **Supplementary Table 3**.

Instead of using GFP as a control for the pulldown, we decided to use a vector expressing TurboID from the mitochondria. These mice express TurboID-Cpt1a from the same AAV and promoter. This should be a better than comparing the data to eGFP. Indeed, we present this data in **Supplementary Figure 2**. We also show the enrichment of Pyruvate Carboxylase (an endogenously biotinylated mitochondrial matrix protein) is identified in both sets of data and is unchanged. The dataset was filtered for proteins with at least 2 peptides and found in at least 2/4 replicates for which ever sample had the highest abundance of the protein, statistically significant or not (**Supplementary Table 2**). A total of 2,578 proteins were identified in one or both of the samples. 300 proteins were significantly enriched in the Rab30 sample, while 217 were enriched in the Cpt1a sample. Those that are not significantly enriched in either Rab30 or Cpt1a are likely non-specifically binding to the beads. Significantly enriched interactors for each TurboID fusion were submitted to the DAVID Bioinformatics Database^{23,24} to define the Gene Ontology terms associated with the potential subcellular localization of these proteins (**Supplementary Fig. 2e**). As anticipated, we found that TurboID-Cpt1a interactors were significantly associated with the mitochondria, while TurboID-Rab30 interactors were more broadly associated with the ER, Golgi apparatus, endosome, cytoplasm, and cell junctions. These data are presented to support the observed distribution of Rab30 in the mouse liver and our live cell imaging data.

4. It is interesting that the authors identify biotinylated peptides for the cytoplasmic domains of plasma membrane receptors such as LRP1 and ASGR1. However, the authors didn't present any detailed information about the MS identification and quantification of these biotinylated peptides including PSMs, FDR and relative quantification as compared with their EGFP control.

We have included the requested information in tabular format (Supplementary Table 3) in the newest version of the manuscript.

References

- 1 Kelly, E. E. *et al.* Rab30 is required for the morphological integrity of the Golgi apparatus. *Biol Cell* **104**, 84-101 (2012). <https://doi.org:10.1111/boc.201100080>
- 2 Zulkefli, K. L. *et al.* A role for Rab30 in retrograde trafficking and maintenance of endosome-TGN organization. *Exp Cell Res* **399**, 112442 (2021). <https://doi.org:10.1016/j.yexcr.2020.112442>

- 3 Larocque, G., La-Borde, P. J., Clarke, N. I., Carter, N. J. & Royle, S. J. Tumor protein D54 defines a new class of intracellular transport vesicles. *The Journal of cell biology* **219** (2020). <https://doi.org:10.1083/jcb.201812044>
- 4 Larocque, G. *et al.* Intracellular nanovesicles mediate $\alpha 5\beta 1$ integrin trafficking during cell migration. *The Journal of cell biology* **220** (2021). <https://doi.org:10.1083/jcb.202009028>
- 5 Larocque, G. & Royle, S. J. Integrating intracellular nanovesicles into integrin trafficking pathways and beyond. *Cellular and Molecular Life Sciences* **79**, 335 (2022). <https://doi.org:10.1007/s00018-022-04371-6>
- 6 Nakajima, K. *et al.* RAB30 regulates PI4KB (phosphatidylinositol 4-kinase beta)-dependent autophagy against group A Streptococcus. *Autophagy* **15**, 466-477 (2019). <https://doi.org:10.1080/15548627.2018.1532260>
- 7 Oda, S. *et al.* Golgi-Resident GTPase Rab30 Promotes the Biogenesis of Pathogen-Containing Autophagosomes. *PLoS one* **11**, e0147061 (2016). <https://doi.org:10.1371/journal.pone.0147061>
- 8 Homma, Y. *et al.* Comprehensive knockout analysis of the Rab family GTPases in epithelial cells. *The Journal of cell biology* **218**, 2035-2050 (2019). <https://doi.org:10.1083/jcb.201810134>
- 9 Xu, Y. *et al.* Hepatocyte ATF3 protects against atherosclerosis by regulating HDL and bile acid metabolism. *Nature metabolism* **3**, 59-74 (2021). <https://doi.org:10.1038/s42255-020-00331-1>
- 10 Xu, X., Park, J. G., So, J. S., Hur, K. Y. & Lee, A. H. Transcriptional regulation of apolipoprotein A-IV by the transcription factor CREBH. *Journal of lipid research* **55**, 850-859 (2014). <https://doi.org:10.1194/jlr.M045104>
- 11 Hanniman, E. A., Lambert, G., Inoue, Y., Gonzalez, F. J. & Sinal, C. J. Apolipoprotein A-IV is regulated by nutritional and metabolic stress: involvement of glucocorticoids, HNF-4 alpha, and PGC-1 alpha. *Journal of lipid research* **47**, 2503-2514 (2006). <https://doi.org:10.1194/jlr.M600303-JLR200>
- 12 Nagasawa, M. *et al.* Identification of a functional peroxisome proliferator-activated receptor (PPAR) response element (PPRE) in the human apolipoprotein A-IV gene. *Biochemical pharmacology* **78**, 523-530 (2009). <https://doi.org:10.1016/j.bcp.2009.05.007>
- 13 VerHague, M. A., Cheng, D., Weinberg, R. B. & Shelness, G. S. Apolipoprotein A-IV expression in mouse liver enhances triglyceride secretion and reduces hepatic lipid content by promoting very low density lipoprotein particle expansion. *Arterioscler Thromb Vasc Biol* **33**, 2501-2508 (2013). <https://doi.org:10.1161/atvbaha.113.301948>
- 14 Lee, J., Choi, J., Scafidi, S. & Wolfgang, M. J. Hepatic Fatty Acid Oxidation Restrains Systemic Catabolism during Starvation. *Cell reports* **16**, 201-212 (2016). <https://doi.org:10.1016/j.celrep.2016.05.062>
- 15 Lee, J. *et al.* Loss of Hepatic Mitochondrial Long-Chain Fatty Acid Oxidation Confers Resistance to Diet-Induced Obesity and Glucose Intolerance. *Cell reports* **20**, 655-667 (2017). <https://doi.org:10.1016/j.celrep.2017.06.080>
- 16 Sarbassov, D. D., Guertin, D. A., Ali, S. M. & Sabatini, D. M. Phosphorylation and regulation of Akt/PKB by the rictor-mTOR complex. *Science (New York, N.Y.)* **307**, 1098-1101 (2005). <https://doi.org:10.1126/science.1106148>
- 17 Hesse, D., Jaschke, A., Chung, B. & Schürmann, A. Trans-Golgi proteins participate in the control of lipid droplet and chylomicron formation. *Bioscience reports* **33**, 1-9 (2013). <https://doi.org:10.1042/bsr20120082>
- 18 Hesse, D. *et al.* Hepatic trans-Golgi action coordinated by the GTPase ARFRP1 is crucial for lipoprotein lipidation and assembly. *Journal of lipid research* **55**, 41-52 (2014). <https://doi.org:10.1194/jlr.M040089>

- 19 Jaschke, A. *et al.* The GTPase ARFRP1 controls the lipidation of chylomicrons in the Golgi of the intestinal epithelium. *Human molecular genetics* **21**, 3128-3142 (2012).
<https://doi.org:10.1093/hmg/dds140>
- 20 Zahn, C. *et al.* Knockout of Arfrp1 leads to disruption of ARF-like1 (ARL1) targeting to the trans-Golgi in mouse embryos and HeLa cells. *Molecular membrane biology* **23**, 475-485 (2006).
<https://doi.org:10.1080/09687860600840100>
- 21 Gillingham, A. K., Sinka, R., Torres, I. L., Lilley, K. S. & Munro, S. Toward a comprehensive map of the effectors of rab GTPases. *Dev Cell* **31**, 358-373 (2014).
<https://doi.org:10.1016/j.devcel.2014.10.007>
- 22 Sinka, R., Gillingham, A. K., Kondylis, V. & Munro, S. Golgi coiled-coil proteins contain multiple binding sites for Rab family G proteins. *The Journal of cell biology* **183**, 607-615 (2008).
<https://doi.org:10.1083/jcb.200808018>
- 23 Huang da, W., Sherman, B. T. & Lempicki, R. A. Systematic and integrative analysis of large gene lists using DAVID bioinformatics resources. *Nat Protoc* **4**, 44-57 (2009).
<https://doi.org:10.1038/nprot.2008.211>
- 24 Sherman, B. T. *et al.* DAVID: a web server for functional enrichment analysis and functional annotation of gene lists (2021 update). *Nucleic acids research* **50**, W216-w221 (2022).
<https://doi.org:10.1093/nar/gkac194>

REVIEWER COMMENTS

Reviewer #1 (Remarks to the Author):

I appreciate the extensive rebuttal by the authors and the detailed answer to my comments. Nonetheless, in my view, the title still does not properly reflect the results obtained in the paper. My suggestion would be to change the title to: Hepatocyte Rab30 deficiency in the context of impaired hepatocyte fatty acid oxidation lower fasting lipemia in mice. This is because no evidence is presented that lipid homeostasis in the hepatocytes is altered. Changes in plasma TG, NEFA and cholesterol are not necessarily caused by events occurring in the liver. Liver TG content was not altered by Rab30 deletion in hepatocyte CPT2 deficient mice.

The legend of figure 6 is incomplete. Please clarify what is shown in figure 6a, b and c?

Reviewer #3 (Remarks to the Author):

Thank you for addressing my comments. The manuscript is much improved. In light of the modest phenotype of the Rab30 single LKO, a less definitive title may be appropriate. I think you can replace "regulates" with a phrase that doesn't paint your work into a corner (facilitates?).

Reviewer #4 (Remarks to the Author):

I don't think the authors properly addressed my comments rather than providing some supplementary tables. Current description is quite general and is not sufficient to draw the discussed conclusion for picking up the proteins as the authors validated. Supplementary Fig. 2d didn't give us the clue how the authors picked up the validated protein from so many changed proteins. The identification and quantification for biotinylated peptides is more questionable. Specifically, for example, I even couldn't find the Lrp1 biotinylated peptide as the authors listed in the main figure in the supplementary table 3. The authors should get support from someone who is well trained for quantitative proteomic analysis, especially for PTM-modified peptide.

REVIEWER COMMENTS

We again thank the reviewers for their comments. Reviewers 1 and 2 suggested a title change.

We would suggest a compromise of “**Rab30 facilitates lipid homeostasis during fasting**”.

Reviewer #1 (Remarks to the Author):

I appreciate the extensive rebuttal by the authors and the detailed answer to my comments. Nonetheless, in my view, the title still does not properly reflect the results obtained in the paper. My suggestion would be to change the title to: Hepatocyte Rab30 deficiency in the context of impaired hepatocyte fatty acid oxidation lower fasting lipemia in mice. This is because no evidence is presented that lipid homeostasis in the hepatocytes is altered. Changes in plasma TG, NEFA and cholesterol are not necessarily caused by events occurring in the liver. Liver TG content was not altered by Rab30 deletion in hepatocyte CPT2 deficient mice.

The legend of figure 6 is incomplete. Please clarify what is shown in figure 6a, b and c?

Thank you for your comment. We have changed the title of the manuscript and hope you find it better suits the results of the paper. We have clarified the legend for Fig. 6. The new figure legend for figure 6 is as follows:

Figure 6. Loss of Rab30 influences fasting circulating triglyceride and cholesterol and suppresses dyslipidemia in fatty acid oxidation deficiency. a Fed and fasting triglyceride (TG), nonesterified fatty acid (NEFA), cholesterol, and β -hydroxybutyrate (β HB) levels in the serum of male Rab30KO and littermate control (Ctrl) mice (n=6-8). **b** Fasting TG, NEFA, cholesterol, and β HB levels in the serum of male Rab30L^{-/-} and control (ff) mice (n=7-8). **c** Fed and fasting TG, NEFA, cholesterol, and β HB levels in the serum of male Cpt2L^{-/-}, DKO, and control (ff;ff) mice (n=7-8). Significant differences (p-value<0.05) were determined by two-tailed unpaired t-test in **a** and **b** and by Tukey's multiple comparison's test following ANOVA in **c**. *, p<0.05; shared letters indicate same significance level.

Reviewer #3 (Remarks to the Author):

Thank you for addressing my comments. The manuscript is much improved. In light of the modest phenotype of the Rab30 single LKO, a less definitive title may be appropriate. I think you can replace "regulates" with a phrase that doesn't paint your work into a corner (facilitates?).

We have changed the wording of the title and replaced the definitive word “regulates” to a more general term “facilitates.”

Reviewer #4 (Remarks to the Author):

I don't think the authors properly addressed my comments rather than providing some supplementary tables. Current description is quite general and is not sufficient to draw the discussed conclusion for picking up the proteins as the authors validated. Supplementary Fig. 2d didn't give us the clue how the authors picked up the validated protein from so many

changed proteins. The identification and quantification for biotinylated peptides is more questionable. Specifically, for example, I even couldn't find the Lrp1 biotinylated peptide as the authors listed in the main figure in the supplementary table 3. The authors should get support from someone who is well trained for quantitative proteomic analysis, especially for PTM-modified peptide.

We apologize that we likely did not fully appreciate the reviewer's comments. The reviewer asked for us to place the proteomics data in a volcano plot. We have now done this and presented it in the Figure. We were attempting to be as rigorous as possible by further defining the site of biotinylation on the proteins identified (biosite). However, the reviewer does not feel that this data adds to the manuscript so we have taken it out and focused solely on the proteins identified and quantified. Therefore, the TurboID experiments are very straightforward.

With the help of our mass spectrometry core facility, we used a label-free quantitation workflow and CHIMERYs post-DDA to quantify the peptide abundances in each pulldown based off of the precursor ion intensity, followed by statistical analysis to find significantly enriched proteins in either the Rab30 or Cpt1a pulldowns (reported in the methods section and what was Supplementary Tables 2 and 7). The dataset identifier is PXD050376. Here are the updated methods with references to the new tables:

***in vivo* TurboID and biotinylated protein enrichment**

3XHA-TurboID-Rab30, TurboID-Cpt1a, or EGFP expressing mice were injected either in the fed state or after a 21hr fast with 24mg/kg bodyweight biotin;^{1,2} a 3hr labeling period was allowed prior to dissection. Protein was extracted from flash frozen liver in lysis buffer with protease and phosphatase inhibitors on ice. Total protein of clarified lysates was quantified by BCA, and 5mg protein was diluted to a final volume of 600µL in dilution buffer (10mM Tris-HCl/150mM NaCl/0.5mM EDTA with protease and phosphatase inhibitors added). Biotinylated peptides were immunoprecipitated on streptavidin magnetic beads (NEB, S1420S) overnight at 4°C, rotating end-over-end. After removing flowthrough, beads were washed 3x in dilution buffer, 3x in 1%SDS, and 3x in 1XPBS, changing tubes every second wash. For western blot analysis of enriched biotinylated proteins depicted in **Supplementary Fig. 2**, bound proteins were eluted by boiling all or a fraction of the beads in sample buffer (Bio-Rad, 1610747) and β-mercaptoethanol for 10mins at 95°C followed by magnetic separation of the beads and isolation of the eluted fraction. For protein identification of streptavidin pulldown samples by mass spectrometry, beads were frozen in 1X PBS prior to elution.

Mass spectrometry analysis of streptavidin pulldown samples following on-bead digestion

100ul of 100mM Triethyl Ammonium Bicarbonate (TEAB, pH 8) was added to the solution containing the streptavidin beads, followed by reduction with 50mM Dithiothreitol at 60°C for 45 minutes and subsequent alkylation with 100 mM Iodoacetamide at room temperature in the dark for 15 minutes. The samples were then digested on the beads by adding 8 ug of Trypsin/LysC (1:50 enzyme:protein) mixed protease (Pierce) and incubating overnight at 37°C and 1000 rpm.

Beads were separated from the digested peptides magnetically. The peptides were further processed to remove detergents and background components using Sera-Mag beads (GE Healthcare GE24152105050350) and standard SP3 peptide cleanup protocols. The TurboID-Rab30 and TurboID-Cpt1a samples were injected into a Thermo Fisher Scientific Orbitrap Fusion Lumos or Q Exactive Plus Orbitrap mass spectrometers, respectively, and eluted over a 90-minute gradient from 2% to 90% acetonitrile containing 0.1% formic acid. Mass spectrometry settings were 120,000 resolution for MS1 precursors and 30,000 for MS2 fragment ions with a 3 second cycle time between precursors. AGC for MS2 was set to 200% (1e5) with a maximum injection time of 54ms and a normalized collision energy of 34.

Raw files obtained from mass spectrometry runs were searched by a label-free quantitation workflow in Proteome Discoverer (Thermo Scientific, version 3.1.0.638) against the Uniprot UP589 *Mus musculus* database using the CHIMERYS identification node prediction model inferys_3.0.0_fragmentation. Oxidation on M and Carbamidomethyl (C) were set by CHIMERYS as dynamic and static modifications, respectively. **Supplementary Table 5** provides the total list of proteins identified across all replicates of TurboID-Rab30 and TurboID-Cpt1a streptavidin pulldowns prior to filtering for proteins with at least 2 peptides, removal of contaminants (i.e., keratin), and normalizing intensities to pyruvate carboxylase levels. We normalized to pyruvate carboxylase, a highly abundant and endogenously biotinylated protein, so that we could better compare the TurboID-Rab30 and TurboID-Cpt1a datasets, as they were run on different mass spectrometers. Additionally, statistically significantly enriched proteins were retained only if the proteins were found in at least 2 of the 4 replicates. **Supplementary Table 2** displays the proteins identified in replicates of TurboID-Rab30 or TurboID-Cpt1a livers after normalization to pyruvate carboxylase and filtering as described and was used to generate the volcano plot in **Fig. 2f**. For both **Supplementary Table 2** and **5**, statistical significance is reported as the adjusted *p*-value using the Benjamini-Hockberg correction for the false discovery rate (FDR). Gene Ontology pathway analysis for cellular component and biological process terms associated with significantly enriched interactors was performed using the DAVID bioinformatics database as cited in the manuscript body.

- 1 Wei, W. *et al.* Cell type-selective secretome profiling in vivo. *Nature chemical biology* **17**, 326-334, doi:10.1038/s41589-020-00698-y (2021).
- 2 Uezu, A. *et al.* Identification of an elaborate complex mediating postsynaptic inhibition. *Science (New York, N.Y.)* **353**, 1123-1129, doi:10.1126/science.aag0821 (2016).

REVIEWERS' COMMENTS

Reviewer #4 (Remarks to the Author):

The authors have addressed my major concerns.